

# Meteorological Analysis of the Forcett-Dunalley Wildfire in 2013 in Tasmania, Australia

Ivana Čavlina Tomašević[1], Paul Fox-Hughes[2], Kevin K. W. Cheung[3*], Višnjica Vučetić[4], Jon Marsden-Smedley[5], Paul J. Beggs[6], Maja Telišman Prtenjak[7]

[1] Croatian Meteorological and Hydrological Service, 10 000 Zagreb, Croatia
[2] Bureau of Meteorology, Hobart, TAS 7000, Australia
[3] School of Emergency Management, Nanjing University of Information Science and Technology, Nanjing, Jiangsu 210044, China
[4] Retired from[1]
[5] Retired, Hobart, TAS 7000, Australia
[6] School of Natural Sciences, Faculty of Science and Engineering, Macquarie University, Sydney, NSW 2109, Australia
[7] Department of Geophysics, Faculty of Science, University of Zagreb, 10 000 Zagreb, Croatia

* *Correspondence to*: Kevin Cheung (kevin.cheung@nuist.edu.cn)

**Abstract.** A major bushfire occurred during January 2013 near the towns Forcett and Dunalley in southeast Tasmania, Australia. Several records were broken by this wildfire, in terms of impacts to eco-systems, infrastructure and lives, as well as the first documented fire storm development in Tasmania in the form of pyrocumulonimbus. The Australian Bureau of Meteorology high-resolution regional reanalysis for Tasmania (BARRA-TA), with 1.5-km spatial resolution, together with in-situ observations, was applied to reconstruct the wildfire event. The antecedent climatic conditions in Tasmania included large increase in fuel load due to abundant rain one to two years before the event, followed by a heatwave during the summer of 2012/13. In the three periods we identified during the event reconstruction, the second period was the most dramatic, in which a low-level jet was directed downslope in southeast Tasmania to accelerate the fire spread. Moreover, spotting of over 3 km was observed, and pyrocumulonimbus developed in this period with lightning up to 13 km from the fire. A cold front crossed the fireground during the third period, and thus played a different role compared with some past extreme fire events in terms of lifting and wind direction change. Our analyses conclude that climatic conditions, synoptic patterns and mesoscale convective environment all contributed to this wildfire event.

## 1 Introduction

Tasmania, the island state of Australia, shares a history of frequent wildfire events with the mainland. Fire plays an important role in Tasmania's ecosystem. Some native flora in Tasmania recovers well from fire or even relies on fire for regeneration (Yospin et al., 2015; French et. al., 2016; Kirkpatrick et al., 2018). For thousands of years fire has been used by Indigenous people for land management purposes, maintaining biodiversity and hunting (Marsden-Smedley, 2014). These practices have been inherited by European settlers and used in 19[th] and most of the 20[th] century (Von Platen et al., 2011). The tradition was gradually abandoned, especially after several disastrous fire seasons and catastrophic wildfires. For example, during fire seasons of 1897-98 and 1933-34 approximately 2 000 000 ha (a third of the state) and 1 000 000 ha, respectively, were burnt. The deadliest wildfire that occurred in Tasmania to date entered the capital city of Hobart, killed 62 people, destroyed 1400 buildings and burnt 250 000 ha, mostly in a single day on 7 February 1967 (Bond et al., 1967; Brotak and Reifsnyder, 1977; Marsden-Smedley, 2014).

The main causes of wildfires in Australia are lightning strikes and human activities, with the number of ignitions caused by lightning strikes increasing in recent decades (Kirkpatric et al., 2018; Styger et al., 2018; Zylstra, 2018). Recent studies suggest that most wildfires in eastern and northern Tasmania to date are still human-caused (Nampak et al., 2021). In the period between 2011 and 2019, ignition causes were classified as accidental (25%), deliberate (11%), planned burn (22%),





undetermined (36%), and lightning induced (7%). However, despite being the least frequent of the ignition sources, lightning was responsible for 53% of the state's total burned land throughout the period (Nampak et al., 2021).

The peak fire weather in Tasmania occurs in late summer and early autumn (Luke and McArthur, 1978), however, recent studies also revealed a springtime peak in fire danger, which appears approximately once every two years (Fox-Hughes, 2008). The most fire-prone area is the southeast of the state (Fox-Hughes, 2008). The average burnt area in Tasmania in the period 2002–2011 was 51 920 ha in 65 wildfires per year. The burnt area escalated in the 2012-13 fire season when 128 wildfires burnt 119 267 ha, most of which was in the Forcett-Dunalley wildfire at the beginning of January 2013. It is worth noting too,

for completeness, that the 2015-16 (Press, 2016) and 2018-19 (Wardlaw, 2021) Tasmanian fire seasons were also markedly more active than average. Mostly, however, the largest fires during these seasons occurred in western Tasmania.

The 2013 Forcett-Dunalley extreme event calls for special attention and gives an opportunity to study the most severe fire weather conditions that can occur in the southeast of Australia. Therefore, the aim of this study is to investigate key weather conditions related to this extreme event. Specifically, to examine antecedent climatic conditions as well as synoptic and

mesoscale weather factors coincident to the event and furthermore, to relate atmospheric processes to fire behavior.

The article is organized as follows: Section 2 presents a detailed description of the Forcett-Dunalley wildfire, Section 3 describes data and methods, Section 4 details observed and modeled atmospheric conditions prior to and in the first 82 hours of the wildfire, and Section 5 provides discussion and a summary.

## 2 Overview of the Forcett-Dunalley wildfire

The Forcett-Dunalley wildfire started from an accidental ignition at 14:00 AEDT (UTC + 11 h) on 3 January 2013 in the rural locality of Forcett, about 30 km east northeast of Hobart, the state's capital (Fig. 1a). The Forcett-Dunalley wildfire was one of four fires ignited accidentally or by lightning on the same day (Fig. 1b, Table 1). Although it was not the largest considering the final burnt area, the wildfire endangered lives and burnt a number of houses in Dunalley, a town located 15 km southeast of Forcett (Fig. 1a). In the afternoon on 4 January 2013 the wildfire produced the fire storm in a form of pyrocumulonimbus

(pyroCb, Fig. 2a) cloud, which is the first clearly documented pyroCb in Tasmania due to its occurrence to the close proximity to weather stations (Fig. 1).

The Forcett-Dunalley wildfire burned for 16 days, from 3 January to 18 January 2013, however it wasn't completely extinguished until 20 March 2013. The wildfire quickly gained global media attention (The Guardian, 2013). It destroyed or damaged 431 properties, including the primary school and police station in Dunalley, and forced people to jump into the sea

to avoid death. One firefighter involved in the intervention lost his life, but not as a direct result from a fire. The wildfire burnt 25 950 ha of native forests, agriculture land, forest plantations, more than 660 km of commercial fencing and killed 10 000 livestock, mainly sheep. The Forcett-Dunalley wildfire had a negative effect on businesses including tourism, livestock farming, agriculture, seafood industries etc. According to estimates, Tasmanian wildfires from January 2013 cost AU$71.9 million in total, before accounting for emergency response and recovery operations costs (Marsden-Smedley, 2014).


Table 1: Summary of five wildfires started from 3 to 5 January 2013 in Tasmania.

| FIRE | LAKE REPULSE | FORCETT-DUNALLEY | GIBLIN RIVER | BICHENO | MONTUMANA |
|---|---|---|---|---|---|
| IGNITION CAUSE | Escaped campfire | Re-ignition campfire | Lightning | Lightning | Lightning |
| IGNITION TIME (AEDT) | 11:30 3.1.2013. | 14:00 3.1.2013. | afternoon 3.1.2013. | . 20:00 3.1.2013. | 08:00 5.1.2013. |
| CONTAINMENT | 18.2.2013. | 18.1.2013. | 22.1.2013. | 9.1.2013. | 20.1.2013. |
| BURNT AREA | 10637 ha | 25950 ha | 45124 ha | 4939 ha | 3158 ha |



| FINAL PERIMETER | 121.5 km | 310 km | 387 km | 41.3 km | 28.5 km |
|---|---|---|---|---|---|

The wildfire occurred in southeast Tasmania (Fig. 1b), along the highly indented coastline with numerous peninsulas, bays and islands. Topography of the region is notable for its undulating terrain, with mean slopes > 10° (Ndalila et al., 2018). The wildfire mostly burned in a southeasterly direction from its ignition location in Forcett. The hills around Forcett and Dunalley, as well as on the nearby Forestier and Tasman Peninsulas southeast of these townships, are approximately 200 m to 400 m high. Similar undulating, but higher elevated terrain is situated northwest of the ignition location with hills between 500 m and 750 m a.s.l. The landscape is dominated by dry and wet *Eucalyptus* forests, and scattered agriculture, with a minor component of *Pinus radiata* plantations (Ndalila et al., 2018).

For the purpose of this study the first 82 hours of the Forcett-Dunalley wildfire will be noted as DUNALLEY 1 to 3. The DUNALLEY 1 period will refer to the first 22 hours of the wildfire, or a period from the re-ignition on 3 January to midday on 4 January, immediately before the escalation in fire activity. The DUNALLEY 2 period will refer to the firestorm that occurred in the afternoon hours on 4 January. The third and final period, DUNALLEY 3, will refer to the change in fire front direction on 5 January and back-burning on 6 January. The time zone used in the wildfire description and further meteorological analysis is Australian Eastern Daylight Time (AEDT), which is UTC + 11 h. The following sections provide the detailed description of denoted burn periods.

### 2.1 Burn period DUNALLEY 1: 14:00 AEDT (3 January)–12:00 AEDT (4 January)

The Forcett-Dunalley wildfire started as a re-ignition from a campfire that had been lit inside an old tree stump on 28 December 2012. In the following days most probably a slow combustion took place within tree roots in very dry soil, until fire reached the surface and wind lifted embers and spread the fire in nearby grass (BoM, 2013a). The official cause of the Forcett-Dunalley wildfire was declared to be an accident.

From the time of the re-ignition at 14:00 AEDT on 3 January near the locality of Forcett, the fire progressed in a southeasterly direction, down a slope of 5°, with an average speed from 6 to 8 m min$^{-1}$. Within an hour the fire burnt 2.5 ha and had a perimeter of 0.7 km. Flame height was up to 5 m, and spotting reached distance to 2.5 km. In the period between 15:00 and 17:30 AEDT fire progressed eastwards increasing spread rate up to 43 m min$^{-1}$. By 17:30 AEDT, or 3.5 hours after the ignition, the fire travelled 5.9 km, burned 506 ha and had a perimeter of 14 km.

During the night between 3 and 4 January the fire activity eased. Although it continued to burn southeast, its overnight spread rate decreased to 2 m min$^{-1}$ and the uncontained southeastern fire front decreased to 12 km in length. The next significant change in fire behavior occurred in the early morning on 4 January. While still slowly progressing south, by 06:45 AEDT the burnt area increased to 973 ha and fire perimeter to 19.6 km. After this time, fire increased its spread rate and intensity and by 12:30 AEDT had a size of 1586 ha with perimeter of 21.7 km (Fig. 3a).

### 2.2 Burn period DUNALLEY 2: 13:00–23:00 AEDT (4 January)

The fire activity escalated after 13:00 AEDT on 4 January 2013. In the following 1.5 hours the fire tripled in size, doubled its perimeter and increased its spread rate to 58.3 m min$^{-1}$. By 14:30 AEDT, 24 hours after the ignition, the wildfire size was 5819 ha with perimeter of 42.8 km. Fire was further progressing southeast, mainly as a high intensity crown fire. Fire intensity, which is determined by fire spread rate, fuel height and fuel load, was estimated at 30 000 kW m$^{-1}$. According to known categorizations the fire intensity between 7000 and 70 000 kW m$^{-1}$ with flame height >15 m classifies a fire as complete crown fire, essentially unstoppable, with firestorm conditions (if reaching an upper limit of fire intensity; Cheney, 1991). Nevertheless, more detailed analysis (Ndalila et al., 2018) found that the fire intensity in the case of the Forcett-Dunalley wildfire during the six-hour period, between 16:00 AEDT and 22:00 AEDT, was even higher and reached 68 571 kW m$^{-1}$. This was estimated for the periods when severe fire weather conditions (according to the Forest Fire Danger Index value) coincided with fire spreading in the wind direction and in the upslope trajectory burning through dry *Eucalyptus* forests. To



put this into perspective, during one of the most catastrophic fires in Australian history, the 2009 Black Saturday fires in the State of Victoria, fire intensity reached 88 000 kW m⁻¹ (Cruz et al., 2012).

The additional proof of extreme fire behavior in the case of the Forcett-Dunalley wildfire is the appearance of violent

pyroconvection (Fig. 2a). Pyroconvection can be manifested as pyrocumulonimbus (pyroCb) cloud, which in Australia had been confirmed in 65 fire cases to 2019 (Ndalila et al., 2019), noting that during the 2019-20 southeastern Australian "super outbreak" an additional 38 pulses of pyroCb activity occurred (Peterson et al., 2021). PyroCb in the case of the Forcett-Dunalley wildfire, which evolution occurred between 13:00 AEDT and 16:00 AEDT on 4 January (Ndalila et al., 2019).

The arrival of the wildfire in Dunalley at 15:25 AEDT was accompanied by an ember storm, which caused multiple spot fires

throughout the town and forced evacuation (The Guardian, 2013). Upon arrival in Dunalley, the fire spread rate was 45 to 50 m min⁻¹. After arriving in Dunalley, fire at one point easily crossed more than 3 km of open water (The Guardian, 2013) and the narrow land neck that connects Forestier Peninsula with the rest of Tasmania before it continued flanking over the peninsula in a southeasterly direction. Along with a third of the Dunalley township, the wildfire also damaged the weather station operated by the Australian Bureau of Meteorology (BoM) (Fig. 2b), strongly affecting the temperature records (BoM, 2013a).

Around 17:30 AEDT, while progressing across the Forestier Peninsula, a probable minor wind change slightly turned fire direction from southeast to south-southeast and pushed the fire towards the already hazard reduced dry forest east of the Murdunna township (Fig. 1a), which explains why there were fewer houses lost in comparison to Dunalley. Between 17:30 AEDT and 20:00 AEDT the fire had an average spread rate of 32.4 m min⁻¹ and during this time period the burnt area increased from 9623 ha to 13 277 ha, and perimeter from 93.6 km to 146.8 km.

By 23:00 AEDT the wildfire reached Eaglehawk Neck, the southernmost locality in the Forestier Peninsula and spotted across the bay onto the Tasman Peninsula (Fig. 1a). This spotting occurred over a distance of approximately 2.5 km. By the end of this progression period, the wildfire had a size of 15 322 ha and a perimeter of 166.9 km (Fig. 3b). Overall, in 11 hours the wildfire progressed approximately 27 km.

### 2.3 Burn period DUNALLEY 3: 5–6 January

Between 01:00 AEDT and 02:00 AEDT on 5 January wind change stopped the wildfire progression towards the southeast and turned the fire front in an east-northeast direction. By 20:30 AEDT the same day, the fire was 19 692 ha in size with a perimeter of 246.6 km. Large-scale back-burning occurred on 6 January northeast of the ignition location in Forcett. The burnt area by 21:00 AEDT on 6 January was 20 981 ha with a perimeter of 269.1 km. The majority of the eventual total fire area burned by this time. Only small additional areas burned until the wildfire was contained 16 days after the ignition, on 18 January 2013

(Fig. 3c).

### 3 Data and methods

### 3.1. Observations

Surface weather conditions during the Forcett-Dunalley wildfire were analyzed using meteorological data from automatic weather stations (AWS) closest to the fire site. The selected meteorological stations included Hobart (51 m.a.s.l.) and Hobart

Airport (4 m.a.s.l.; Fig. 1a). The meteorological data from 3 to 5 January 2013 included 30-minute (on the hour and half hour) air pressure, air temperature, relative humidity, precipitation amount, and mean and maximum wind speed and direction. Mean and maximum wind speeds are 10-minute values available every half hour. Meteorological stations are operated by the BoM. Air temperature and rainfall patterns preceding the Forcett-Dunalley wildfire were accessed using the climate maps over different timescales across Australia and single states, including Tasmania available from the BoM

(http://www.bom.gov.au/climate/, last accessed 20 October 2023).


### 3.2. Fire danger rating

In Australia the Forest Fire Danger Index (FFDI) and the Grassland Fire Danger Index (GFDI) were routinely used to quantify fire weather risk at the time of the Forcett-Dunalley fire. These indices were defined by McArthur (Luke and McArthur, 1978) in the late 1960s to assist in estimation of fire behavior related to the weather. FFDI is calculated using the McArthur Mark 5

Forest Meter (McArthur, 1967) with the inputs of air temperature, relative humidity, average 10-minute wind speed and the drought factor, a measure of the state of the fuel.

The drought factor (DF) is a function of daily rainfall and the period of time elapsed since the last rain (Finkele et al., 2006). It is calculated according to the Griffiths (1999) formulation and, in Tasmania, using the Soil Dryness Index to estimate the deficit of the soil moisture (Mount, 1972). The FFDI calculation assumes a fixed value of 12.5 tons per hectare of fuel load

and it does not take into account the slope of the landscape. In Tasmania, as in other Australian states, forest fire danger is numerically represented by FFDI and in terms of categories of fire activity as (Forecast Fire Danger Rating) FFDR, which assists fire agencies in determining the possible fire behavior and the difficulty to control a fire (Table 2). The FFDI vales have been calculated from selected AWS closest to the wildfire site (Hobart, Hobart Airport, Dunalley, and Campania, Fig. 1). FFDI and FFDR are calculated on a 1-minute and 1-hour basis from 05:00 AEDT to 23:00 AEDT on 4 January 2013, or the period

of the most extreme wildfire activity. Two periods of data were excluded from the analysis. The first is from Hobart station from 16:35 AEDT to 17:58 AEDT when the wet bulb reservoir dried out, which affected the humidity observations. The second period is from 16:19 AEDT to 18:06 AEDT at Dunalley station when the Forcett-Dunalley wildfire burnt in the close vicinity of instruments, leading to highly variable air temperature.

**Table 2. Forest Fire Danger Index (FFDI) values and corresponding Forest Fire Danger Rating (FFDR) and description of difficulty of suppression (Lucas et al., 2007; Vercoe, 2003).**

| Forest Fire Danger Index (FFDI) | Forest Fire Danger Rating (FFDR) | Difficulty of suppression |
|---|---|---|
| 0–11 | Low-Moderate | Fires easily suppressed with hand tools or fire usually suppressed with hand tools and easily suppressed with bulldozers. |
| 12–24 | High | Fire generally controlled with bulldozers working along the flanks to pinch the head out under favourable conditions. Back burning may fail due to spotting. |
| 25–49 | Very high | Initial attack generally fails but may succeed in some circumstances. Back burning will fail due to spotting. Burning-out should be avoided. |
| 50–74 | Severe | Fire suppression virtually impossible on any part of the fire line due to the potential for extreme and sudden changes in fire behaviour. Any suppression actions such as burning out will only increase fire behaviour and the area burnt. |
| 75–99 | Extreme | ??? |
| 100 or higher | Catastrophic | ??? |

### 3.3. Synoptic charts

Synoptic analysis was based on analysis from the National Meteorological and Oceanographic Centre (NMOC) available at

the BoM web page (http://www.bom.gov.au/australia/charts/archive/, last accessed on 20 October 2023). The data included mean sea level pressure analysis for the period from 3 to 8 January 2013, available at 6-hour intervals at synoptic hours for each day.



### 3.4. BARRA reanalysis

The BoM's atmospheric high-resolution Regional Reanalysis for Australia (BARRA) is atmospheric regional reanalysis covering Australia, New Zealand, Southeast Asia and south to the Antarctic ice edge (Su et al., 2019). The BARRA reanalysis includes the 29-year period from 1990 to 2018 and involves two suites. The first suite is regional (identified as BARRA-R) at approximately 12 km resolution, covering the entire Australian region. The second suite consists of sub-areas covering the major cities, including the one centered around Tasmania (BARRA-TA) and dynamically downscaled at 1.5 km resolution. The reanalysis is produced in two steps. In the first step for initialization and boundary conditions BARRA-R uses the European Center for Medium-Range Weather Forecasts (ECMWF) coarse-scale (~ 80 km) global atmospheric reanalysis (ERA-Interim; Dee at al., 2011). BARRA-R further combines conventional observations and short model forecasts to provide the best representation of the atmosphere at approximately 12 km horizontal resolution with 70 vertical levels extending up to 80 km into the atmosphere (50 levels up to 18 km and 20 model levels above 20 km). The second step includes a convective-scale downscaling model over Tasmania. This downscaling model has 1.5 km horizontal resolution and 70 vertical levels up to 40 km in the atmosphere. The sub-domain for Tasmania stretches from 142.5°E to 150.5055°E and from 39.1555°S to 46.0°S (Fig. 4). Both reanalysis and downscaled suite for Tasmania were run four times a day, with data covering a 24-hour period. BARRA-TA produced hourly outputs of meteorological variables (BoM, 2018). BARRA-R and its subdomains, including BARRA-TA, give a realistic reproduction of weather conditions at and near surface and provide an opportunity to analyze past weather, including extreme events, and have various implications for users from a range of fields such as in fire management (e.g., Fox-Hughes et al., 2022). More details on the BARRA reanalysis and evaluation can be found in Su et al. (2019). BARRA reanalysis data are available online at: https://dapds00.nci.org.au/thredds/catalog/cj37/BARRA/BARRA_TA/v1/forecast/catalog.html (last accessed on 23 October 2023).

## 4 Results

### 4.1. Antecedent conditions and fire danger rating

Long-term antecedent conditions in the decade prior to the Forcett-Dunalley wildfire (from hereafter Dunalley wildfire) in Tasmania are characterized as drier and warmer than average. From 2003 to 2012, total rainfall in the Dunalley area was very much below average comparing to 10-year periods from 1900 to 2009 In the years immediately prior to 2013, rainfall deficiencies were common but some periods of higher rainfall did occur. For example, the total rainfall in the eastern part of Tasmania in 2009 and 2011 was above or very much above average (BoM, 2013a). These infrequent, yet relatively wet, conditions most likely contributed to vegetation growth in the Dunalley area before the wildfire in January 2013.

Short-term antecedent conditions included total rainfall below average (Fig. 5a) and air temperature above average (Fig. 5b) for most of the Australian continent in months prior to the Dunalley wildfire. From October to December 2012, rainfall deficiency is evident for most of eastern Australia, including Tasmania, where the total 3-month rainfall in the Dunalley area was below average which contributed to drying out fuels immediately prior to the wildfire.

Maximum daytime temperatures during the last 3 months of 2012 were very much above average (Fig. 5b). The culmination of the prolonged warmer than average period was an extreme heatwave that occurred between 25 December 2012 and 19 January 2013. This heatwave was characterized by its unusual spatial extent and duration of high air temperature. A new record for the hottest day for Australia as a whole was set on 7 January 2013, with an averaged maximum temperature of 40.3°C (previous record of 40.17°C was set on 21 December 1972, and the record was broken again with 41.88°C on 18 December 2019; BoM, 2013b). Furthermore, the area-averaged temperature for Australia exceeded 39°C for seven consecutive days, from 2 to 8 January 2013. To put this into perspective, until 2013 there had been only 21 days in 102 years when the averaged maximum air temperature exceeded 39°C. Overall, January 2013 was the hottest month on record in Australia with both mean





and maximum air temperature setting records of 36.92°C and 39.68°C, respectively, which is 2.28°C and 1.77°C above the

average from the climatological period 1961–1990 (BoM, 2013b).

The peak of the heatwave in Tasmania occurred on 3 and 4 January 2013, the first two days of the Dunalley wildfire. The maximum air temperature on both days was 12°C above average (Fig. 5c), with the minimum air temperature 8°C above average on the morning of 4 January 2013 (Fig. 5d).

The aforementioned antecedent conditions led to the increase in the FFDI and FFDR in Tasmania at the beginning of January

2013. According to 1-hour FFDI values, Hobart and Campania stations recorded a catastrophic fire danger rating in the afternoon on 4 January 2013 (Table 3). The catastrophic rating occurred at 15:00 AEDT, coinciding with the pyroCumulonimbus development within the DUNALLEY 2 period.

**Table 3. Forest Fire Danger Rating (FFDR) and Forest Fire Danger Index (FFDI) at Hobart, Hobart Airport, Dunalley and Campania for the period from 05:00 AEDT to 23:00 AEDT on 4 January 2013. Data outages occurred at Hobart and Dunalley**
**stations between 17:00 and 18:00 AEDT. Missing data at Dunalley station refer to times when wildfire burnt in close proximity to the automatic weather station.**

| | HOBART | | HOBART AIRPORT | | DUNALLEY | | CAMPANIA | |
|---|---|---|---|---|---|---|---|---|
| Time (AEDT) | FFDR | FFDI | FFDR | FFDI | FFDR | FFDI | FFDR | FFDI |
| 05:00 | Very High | 25 | Low-Moderate | 9 | Low-Moderate | 5 | High | 14 |
| 06:00 | Very High | 26 | Low-Moderate | 7 | Low-Moderate | 6 | High | 14 |
| 07:00 | Very High | 29 | Low-Moderate | 10 | Low-Moderate | 7 | Low-Moderate | 11 |
| 08:00 | Very High | 32 | High | 12 | Low-Moderate | 9 | High | 12 |
| 09:00 | Very High | 31 | High | 14 | Low-Moderate | 10 | High | 17 |
| 10:00 | Very High | 30 | High | 15 | Low-Moderate | 9 | High | 21 |
| 11:00 | Very High | 29 | Very High | 25 | Low-Moderate | 9 | Very High | 36 |
| 12:00 | Severe | 74 | Very High | 39 | High | 20 | Severe | 57 |
| 13:00 | Severe | 72 | Extreme | 93 | Very High | 25 | Extreme | 77 |
| 14:00 | Extreme | 77 | Severe | 66 | Severe | 56 | Extreme | 75 |
| 15:00 | Catastrophic | 112 | Severe | 69 | Severe | 72 | Catastrophic | 112 |
| 16:00 | Extreme | 81 | Extreme | 77 | Severe | 70 | Extreme | 85 |
| 17:00 | - | n/a | Extreme | 76 | - | - | Extreme | 88 |
| 18:00 | Severe | 54 | Severe | 56 | - | - | Severe | 53 |
| 19:00 | Very High | 45 | Severe | 56 | Severe | 63 | Very High | 46 |
| 20:00 | Very High | 33 | Very High | 39 | Very High | 38 | Very High | 38 |
| 21:00 | Very High | 30 | High | 22 | Very High | 35 | Very High | 25 |
| 22:00 | High | 22 | Very High | 28 | High | 21 | High | 19 |
| 23:00 | Very High | 25 | Very High | 26 | Very High | 26 | High | 21 |

### 4.2. Surface synoptic conditions

Important fire weather synoptic conditions in the case of the Dunalley wildfire included an anticyclone situated over the Tasman Sea, northeast of Tasmania, and an approaching cold front, west of the state. In days prior to ignition, from 1 to 3

January 2013, the anticyclone moved across the Great Australian Bight and over southeast Australia towards the Tasman Sea (Fig. 6a), i.e., from the northwest to the northeast of Tasmania. The pressure difference between the center of the anticyclone and the cyclone in southwest of the state was very high on 3 January (1018 hPa in contrast to 972 hPa; Fig. 6a). This directed a strong north to northwesterly flow of hot and dry air from the central Australian mainland towards the southeast of the continent and over Tasmania. On the evening on 3 January (the DUNALLEY 1 period) a cloud band stretching from Western

Australia to Tasmania (not shown) reached the southwest and eventually southeast coasts of Tasmania, bringing thunderstorms, yet with rainfall totals below 1 mm in the 24 hours to 09:00 AEDT on 4 January (BoM, 2013a). Lightning had been detected in the area of Forcett and Dunalley, however, no fires were ignited there (although fires were started elsewhere in Tasmania





by dry lightning storms). This cloud band was caused by the line of convergence which moved from Western Australia towards the southeast.

The anticyclone over the Tasman Sea persisted in its position and strength throughout 3 and 4 January, i.e., at the time of the wildfire's ignition on 3 January (the DUNALLEY 1 period) and during the peak of pyroconvection on 4 January (the DUNALLEY 2 period) with central air pressure ranging from 1017 hPa to 1021 hPa. Meanwhile, a cold front was progressing steadily eastward from Western Australia, following the line of convergence which stretched from the southern Australian mainland towards the northwest of Tasmania on 4 January (Fig. 6b, the DUNALLEY 2 period).

The cold front reached Tasmania by the end of the DUNALLEY 2 period and crossed it in the night between 4 January and 5 January (Fig. 6c), in the first hours of the DUNALLEY 3 period. A cold front was associated with a thermal trough over Tasmania on 5 January 2013 (Fig. 6c) which formed due to the hot Tasmanian landmass. The cold front passage was accompanied by a wind direction change, from northwesterly to southwesterly and southeasterly, and isolated showers across western Tasmania which did not bring any significant rainfall overnight or in the morning on 5 January (BoM, 2013a). The

cold front passage was followed by a ridge of high pressure (Fig. 6c), which remained over Tasmania until 7 January 2013.

### 4.3. Observations

Mean sea level pressure observations reflect the synoptic analysis. Air pressure continually dropped from 3 January (the DUNALLEY 1 period) until the peak pyroCb activity on 4 January (the DUNALLEY 2 period). The drop occurred from 1013.5 hPa to the minimum value of 997.3 hPa at 16:00 AEDT on 4 January (Fig. 7a), coinciding with the most significant

period of pyroconvection. From 5 January 2013 (the beginning of DUNALLEY 3 period) air pressure increased and in just 24 hours reached 1022 hPa (by 6 January 2013; Fig. 7a). The automatic weather station at Hobart Airport confirms the same pattern (not shown).

Temperature records were set on 4 January 2013 (the DUNALLEY 2 period). With 41.8°C Hobart station recorded its highest air temperature in 120 years (BoM, 2013a), which is also an all-time record for southern Tasmania. On the same day, Hobart

and Hobart Airport recorded their highest minimum air temperature in January. On the night between 3 and 4 January 2013 (the DUNALLEY 1 period) air temperature did not drop below 20°C (Fig. 7c).

High surface air temperature was accompanied by low relative humidity. At the ignition time on 3 January, relative humidity dropped to 14 % in Hobart and at Hobart Airport (the DUNALLEY 1 period). Expected overnight increase in relative humidity did not occur, although there were disturbances over Tasmania in the night between 3 and 4 January that did not bring any rain

in the area of Forcett and Dunalley. At peak pyroconvection in the early afternoon on 4 January (the DUNALLEY 2 period) relative humidity dropped to its lowest values in this case with only 11 % at Hobart Airport and 12 % in Hobart, both at 15:30 AEDT (Fig. 7c). Therefore, as it might be expected, both surface air temperature and relative humidity confirm the severe hot and dry weather conditions during the first 24 h of ignition. The considerable change in weather conditions occurred around midnight between 4 and 5 January (between DUNALLEY 2 and 3 periods). In only 1.5 hours (from 23:00 AEDT to 00:30

AEDT) air temperature dropped from 31.1°C to 17.9°C and relative humidity increased from 24 % to 81 %.

Another significant feature of this event was a strong and gusty northwesterly wind that peaked on two occasions – immediately after the ignition (the DUNALLEY 1 period) and during the firestorm (the DUNALLEY 2 period). The strongest peak at Hobart (Fig. 6b) and Hobart Airport (not shown) stations occurred right after the ignition. Maximum 10-min wind speed and wind gust of the northwesterly in Hobart was recorded on 3 January at 21:00 AEDT, 15.4 m s$^{-1}$ and 24.7 m s$^{-1}$, respectively.

The maximum wind speed and wind gust in the period of pyroconvection (the DUNALLEY 2 period) was 13.9 m s$^{-1}$ with gusts of 23.1 m s$^{-1}$ at 14:00 AEDT. Both Hobart and Hobart Airport stations recorded a sharp change in wind direction right at midnight on 5 January, indicating the cold front passage (the DUNALLEY 3 period). While wind direction at Hobart Airport (not shown) turned from northwesterly to southwesterly, in Hobart wind changed to southeasterly (Fig. 7d).





Wind speed and direction together with air and dew-point temperature from radiosonde measurements at Hobart Airport
presented as vertical profiles reveal details of vertical conditions in the atmosphere. Two soundings prior to the pyroconvection
(at 23:00 AEDT on 3 January and 11:00 AEDT on 4 January; the DUNALLEY 1 period) present the appearance of a low-
level jet (LLJ) in the lower troposphere (Fig. 8a). LLJ is a significant wind speed maximum (> 12 m s⁻¹) up to 1500 m height
with pronounced wind speed decrease above and up to 3000 m height (Byram, 1954). The wind speed of the night-time LLJ
was 28 m s⁻¹ and of the one preceding the pyroconvection was 24.2 m s⁻¹, which is for both cases, according to previous
research (Bonner, 1968), defined as the strongest LLJ of criterion 3. The wind speed profile also reveals increase of upper-
level wind and the jet stream appearance at the tropopause by the end of DUNALLEY 1 and 2 periods. The jet speed increased
to a maximum of 44.2 m s⁻¹ around 11 km height at 23:00 AEDT on 4 January, right before the cold front passage. The wind
in the pre-frontal air mass on the day of the pyroconvection had northwest direction throughout the whole troposphere (Fig.
8b). Air and dew point temperature vertical profiles reveal dry mid-level air and two temperature inversions up to 2 km height
right before the ignition at 10:00 AEDT on 3 January (not shown).

**4.4. BARRA reanalysis**

**4.4.1 Intense heat and severe ground conditions**

The reanalysis data corroborate the surface synoptic pressure analysis and AWS data from Hobart and Hobart Airport. BARRA
depicts the strong pressure gradient over Tasmania at the time of the wildfire's ignition on 3 January (Fig. 9a; the DUNALLEY
1 period). The pressure gradient from the north to south of the state, approximately over 300 km distance varied from 1013
hPa to 1002 hPa, with the lowest mean sea level pressure in close proximity to the wildfire's location. The pressure gradient
eased towards the evening on 3 January and the following day, with the minimum mean sea level pressure at the fire ground
around the time of pyroconvection (Fig. 9b the DUNALLEY 2 period). According to the BARRA reanalysis, the minimum
mean sea level pressure at Hobart was between 997 hPa and 1002 hPa from 15:00 AEDT to 21:00 AEDT on 4 January, which
corresponds well with to the AWS measurements (Fig. 7a). As the cold front crossed Tasmania overnight on 4-5 January the
mean sea level pressure in the area increased to between 1007 and 1010 hPa (Fig. 9c, the DUNALLEY 3 period).
Extreme surface weather conditions are illustrated by the highest skin (surface) and air temperature and lowest relative
humidity immediately before and during the first 34 hours of ignition (DUNALLEY 1 and 2 periods). Severe conditions on
3 January, right prior to the ignition, were marked with skin temperature between 32 and 34°C (Fig. 10a), air temperature
between 30 and 34°C and relative humidity below 10 % (Fig. 10c, e). On 4 January, weather conditions became even more
extreme. Statewide skin temperature exceeded 30°C (Fig. 10b) while relative humidity remained below 10 % (Fig. 10f). The
Dunalley area again stood out with the highest maximum temperature values of the day. In the hours prior to the
pyroconvection, skin temperature exceeded 40°C (Fig. 10b) while air temperature was between 36 and 38°C (Fig. 10d). The
fact that the BARRA reanalysis slightly underestimated measured values highlights the severity of this case.
The reanalysis confirmed warm surface conditions in the night between 3 and 4 January when air temperature in the area of
Hobart and Dunalley did not drop under 20°C and relative humidity remained below 20 % (not shown). The change in broader
weather conditions after the cold front passage (DUNALLEY 3 period) which decreased temperature and increased relative
humidity was also evident from the reanalysis data (Figs. 11a, b).
After the ignition the wildfire was driven by the northwesterly airflow which was persistent in the Dunalley area during the
DUNALLEY 1 and DUNALLEY 2 periods (from 14:00 AEDT on 3 January to 23:00 AEDT on 4 January). According to
BARRA, westerly to northwesterly wind in the area of the wildfire was strong immediately before and at the ignition time
with mean speed between 8 and 12 m s⁻¹ and wind gusts up to 16 m s⁻¹. At some locations, mostly upwind and further from
the coast, wind gusts at narrow swathes of stronger, gusty wind up to 24 m s⁻¹ (Figs. 12a, b). Decrease in wind speed occurred
throughout the night between 3 and 4 January, with mean speed falling to below 8 m s⁻¹ (not shown). A rapid increase in wind
speed to 16 m s⁻¹ and wind gusts up to 24 m s⁻¹ is evident again between 09:00 AEDT and 15:00 AEDT on 4 January, or by





the end of DUNALLEY 1, and in the first hours of DUNALLEY 2 period (Figs. 12c, d). This strong and persistent northwesterly wind, perpendicular to the coast, coincided with an increase in fire activity in the morning of 4 January and immediately before the pyroconvection.

By the time of the peak pyroconvection, wind gusts maintained strong to gale force upwind of the fire location (Figs. 12c, d).

However, in the downwind area, over the Tasman Peninsula and further offshore, wind slightly eased in strength. This is due to an approaching line of convergence, the edge of which reached the Tasman Peninsula by the time of peak pyroconvection. The line of convergence corresponds to a lower wind zone offshore to the southeast of Tasmania. According to BARRA, the convergence line crossed the wildfire's area around 20:00 AEDT on 4 January (not shown). The line of convergence was followed by the cold front passage at the end of DUNALLEY 2 period. This is evident in a further easing in wind speed in the

broader area and abrupt change in wind direction to southwest during the first hours of DUNALLEY 3 period (Figs. 12e, f). The timing of the cold front passage over Tasmania is represented accurately when compared to synoptic analysis.

### 4.4.2 Severe upper-level conditions and cold front

The upper-level evolution of air temperature, relative humidity and wind provides more evidence on severity of fire weather in this case. Warm and dry conditions at 850 hPa were persistent throughout the DUNALLEY 1 and 2 periods. At the ignition

time at 14:00 AEDT on 3 January, air temperature at 850 hPa over Tasmania was between 16 and 19°C (not shown). Moving forward towards the evening, air temperature did not drop. On the contrary, upper-level conditions became progressively more severe, with air temperature eventually reaching the maximum of 25°C in the night between 3 and 4 January (DUNALLEY 1 period; Fig. 13a). This very warm air remained and covered the wildfire's area for a period of 24 hours, between 22:00 AEDT on 3 January (DUNALLEY 1 period) until 22:00 AEDT on 4 January (end of DUNALLEY 2 period). Extreme warmth at 850

hPa was also followed by reduction in relative humidity which remained between 20 and 40 % during the first 34 hours of ignition or during DUNALLEY 1 and 2 periods (Fig. 13b). The warm and dry air mass covered the area of the state until the cold front crossed Tasmania and cool and moist air advected from the southwest on the night on 5 January (DUNALLEY 3 period; Figs. 13c, d).

Northwesterly wind dominated upper-level air flow over Tasmania during all burn periods of Dunalley wildfire (DUNALLEY

1 to DUNALLEY 3). Although at the ignition time (at 14:00 AEDT on 3 January) airflow was mostly southwesterly to westerly, significant direction shift to northwest occurred at 06:00 AEDT on 4 January (Figs. 14a, b). A shift in wind direction started at the 850 hPa and 500 hPa levels and by the end of DUNALLEY 1 period extended up to 300 hPa (at 11:00 AEDT on 4 January). While the northwesterly wind expanded in coverage by the end of DUNALLEY 1 period, it also increased in speed. This is especially evident during the DUNALLEY 2 period at higher altitudes of 500 hPa and 300 hPa. At 850 hPa

northwesterly wind eased in strength from 10:00 AEDT to 18:00 AEDT, which includes the pyroconvection period. By the end of DUNALLEY 2, wind speed further increased at the upper levels of 500 hPa and 300 hPa and by the end of DUNALLEY 3 period reached maximum speed of 40 m s$^{-1}$ (Fig. 14c). Meanwhile wind speed at 850 hPa eased and shifted to southwesterly with the approaching cold front.

Vertical cross sections up to the top (300 hPa) of the troposphere are derived along the 120 km line (Fig. 1) that extends from

the interior of Tasmania, through the Dunalley wildfire location and over the Tasman Sea following the northwesterly airflow (Fig. 4) and also being perpendicular to the Tasmanian Sea coast.

Vertical cross sections of wind speed reveal the upper-level divergence at the ignition time (the DUNALLEY 1 period) which was followed by the northwesterly airflow throughout the troposphere (DUNALLEY 1 and 2 periods) and gradual strengthening of the upper-level jet during the cold front passage (DUNALLEY 2 and 3 periods). Divergence at the ignition

time occurred around 600 hPa and divided the slower northwesterly air flow below and more accelerated southwest flow above (Fig. 15a). The maximum wind speed in the layer of southwesterly flow was between 25 and 27.5 m s$^{-1}$, which corresponds to





radiosonde measurements obtained a few hours before the ignition at Hobart Airport (Fig. 8a, b). While the lower northwesterly air flow was dry (originated from Australian mainland), the upper-level air flow coming from the southwest was moist, which can be seen in relative humidity increase above 500 hPa after the ignition and during the night between 3 and 4 January (Fig.

15b). Moving forward in time during the DUNALLEY 1 period, southwesterly air flow eased and the direction of the wind aloft gradually aligned with the wind at lower levels (not shown). At 06:00 AEDT on 4 January the northwesterly air flow extended throughout the entire troposphere and remained persistent until the cold front passage at the end of DUNALLEY 2 period (Fig. 15c). From the time of wind direction alignment, upper-level northwesterly wind accelerated. This acceleration was more evident at upper levels, while at lower levels wind speed remained largely constant (Fig. 15c, e). Immediately prior

to the pyroconvection, horizontal wind speed was between 2.5 and 12.5 m s$^{-1}$ up to 900 hPa and between 15 and 17.5 m s$^{-1}$ up to 850 hPa (Fig. 15c). The horizontal wind speed further eased during the period of pyroconvection up to 600 hPa (not shown). Significant change in upper-level conditions was evident by the end of the DUNALLEY 2 and the first hours of DUNALLEY 3 periods. The cross section of the troposphere was again divided with two airflows – the area of low wind up to 850 hPa and the strong jet stream above (Fig. 15e). While the cold front affected the wind at lower levels, decreasing its speed and turning

direction to southwesterly, the upper-level northwesterly wind peaked. The northwesterly airflow accelerated and by the end of DUNALLEY 3 period (at 05:00 AEDT on 5 January) reached the maximum speed of 45 m s$^{-1}$ in a deep layer between 600 hPa and 300 hPa.

Vertical cross sections of potential temperature reveal vertical mixing, which was progressively getting stronger from the ignition (Fig. 15b; the DUNALLEY 1 period) until pyroconvection (Fig. 15d; the DUNALLEY 2 period). The lack of any

significant potential temperature gradient in the layer up to 600 hPa was accompanied by low relative humidity, which by the time of pyroconvection dropped under 30% (Fig. 15d). Cross sections of relative humidity also corroborate the previous synoptic analysis and confirm the advection of moisture over Tasmania after the cold front passage overnight on 5 January 2013, when the layer of moist air extended throughout the entire troposphere (Fig. 15f).

## 5 Discussion and conclusions

The Forcett-Dunalley fire in January 2013 was one of the most extreme wildfires in Tasmanian recent history given the size, damage caused, rapid spread and unexpected fire behaviour which included generation of a firestorm, the first clearly documented pyroCb in Tasmania to date. This research aims to answer several questions on weather conditions preceding this wildfire event as well as those related to the fast fire growth and the development of a firestorm in the first 82 hours following ignition, also known as fire propagation periods DUNALLEY 1 to 3.

In the decade prior to the wildfire, long-term weather conditions are classified as a combination of frequent warm and dry weather with occasional periods of above average rainfall (BoM, 2013). Although infrequent, these relatively wet periods of above average rainfall contributed to vegetation growth in the area of Forcett and Dunalley, thus increasing fuel load. This is exactly what happened one year before the wildfire, in 2011. This abundance of vegetation dried out in the months immediately prior to the wildfire due to dry and warmer than average conditions. Conditions favourable to wildfires culminated as an

extreme heatwave at the end of December 2012 and during most of January 2013. The heatwave characterized by its unusual temporal and spatial extent resulted in the hottest month on record in Australia, including Tasmania, and its peak coincided with the ignition of the Dunalley wildfire. Specifically, the peak of the heatwave occurred during the first 48 hours of the wildfire, with a maximum air temperature anomaly 12°C above average and minimum air temperature 8°C above average. The heatwave led to an all-time air temperature record for southern Tasmania with the Hobart station measuring its highest air

temperature in 120 years. These extreme conditions caused an increase in fire danger, which reached catastrophic rating in the afternoon on 4 January 2013, coinciding with the pyroconvection. The catastrophic fire danger rating highlighted conditions on that day as 'the near worst possible fire conditions that are likely to be experienced in Australia' (Lucas, 2010). These


exceptional circumstances reflect the state across the rest of Tasmania affected by multiple wildfires which burned in a single day twice the area typically burnt in an entire season in Tasmania.

The severe antecedent conditions were combined with favourable fire weather which included a strong surface pressure gradient in the area caused by a synoptic configuration which consisted of (1) the anticyclone situated in the Tasman Sea, northeast of Tasmania and (2) an approaching cold front from the west. The adjustment of these patterns resulted in the strong northwesterly airflow which brought the intense heat, i.e., hot and dry air from the central interior of the Australian mainland. The aforementioned synoptic patterns are known to produce the most severe fire weather conditions in southeast Australia

(e.g., Foley, 1947; Cheney, 1976; Mills, 2005; Fox-Hughes, 2012; Tomašević et al, 2022; Fox-Hughes, 2023). Historical wildfires in Victoria, New South Wales and Tasmania are often related to this particular feature of fire weather – dry summertime cold fronts preceded by noticeable anticyclone (e.g., Reeder and Smith, 1992). As mentioned, this synoptic configuration directs dry and heated air from the continent interior with prefrontal wind of northerly directions and has been found to have a strong correlation with high fire risk and the appearance of the most catastrophic wildfires in Australia. For

instance, weather conditions that included persistent anticyclone followed by a dry cold front are found in 24 large wildfires in southeast Australia in the period from 1962 to 2003 (Mills, 2005; Long, 2006; Reeder, 2015).

According to the presented meteorological analysis, the most significant fire behaviour in the case of Dunalley wildfire can be explained as follows:

1)      Surface synoptic conditions coinciding with the wildfire ignition are well known to contribute to fire risk and

disastrous fire weather conditions, bringing the hot and dry air from the continental interior towards Tasmania. The northwesterly wind determined the fire activity during the DUNALLEY 1 period. A strong surface northwesterly air flow with occasional gale force gusts at the ignition time pushed the wildfire in a southeast direction, towards the township of Dunalley. Strong wind could easily lift embers (e.g. Cruz et al., 2012; Sharples et al. 2016; Wilke et al. 2022) and as the wind speed increased until the midnight on 4 January so did the wildfire spread rate (mid DUNALLEY 1 period).

2)      The ease in fire activity during the night between 3 and 4 January (mid DUNALLEY 1 period) was primarily caused by a decrease in wind speed. In spite of high overnight air temperature (> 20°C up to 850 hPa) and low relative humidity (< 30%) which would favour the fire activity, the weak northeasterly wind in the first hours of 4 January decreased and almost stopped any fire spread and progression. While the low relative humidity persisted in the lower atmosphere, the existence of high relative humidity above the layer of 500 hPa was related to the cloud band stretching from Western Australia to Tasmania,

which brought thunderstorms, yet not in the Dunalley area. It is well known that atmospheric moisture has a major influence on fire danger and behaviour (Potter, 2012a) and that low relative humidity increases fire size and number of fires (e.g., Follweiler, 1929; Lloyd, 1932; Robin and Wilson, 1958; Potter, 1996). Also, the lack of the overnight process of cooling and fuel moisture recovery which reduces dead fuel moisture in the area of the wildfire is found to drastically enhance fire activity (e.g., Gisborne, 1927; Čavlina Tomašević et al., 2022). This did not happen in the case of Dunalley wildfire. On the contrary,

the general ease in wind speed caused the decrease of fire activity in spite of warm and dry overnight conditions. However, those unusual overnight conditions did set the course for dramatic increase in fire activity during the following day.

3)      The re-activation and escalation of the fire activity in the morning and during the day on 4 January (the end of DUNALLEY 1 and DUNALLEY 2 period) coincided with the most extreme surface weather conditions which included maximum air temperature exceeding 40°C, relative humidity dropping to 10 % and northwesterly wind increasing, which

altogether subsequently produced the catastrophic fire danger rating (FFDI of 112) – a rarity in Tasmania. In the afternoon on 4 January 2013 (the DUNALLEY 2 period) fire activity escalated and Dunalley wildfire blew up into a pyroCb storm. Firestorms in the form of pyroCb clouds are found to have a significant influence on fire behaviour. For instance, firestorms amplify fire spread and burn rate (Banta et al., 1992; Fromm et al., 2006; Trentmann et al., 2006; Rosenfeld et al., 2007; Fromm et al., 2012) and enhance spotting causing lifting of burning embers which could be carried into unburned ground and results

in new ignitions (Koo et al., 2010). Both of these phenomena occurred during the Dunalley wildfire. Right before and while


burning through Dunalley township the wildfire increased its speed rate up to 58 m min$^{-1}$ and at one point crossed 3 km of open water and started new ignitions on the nearby Tasman Peninsula. The Dunalley firestorm also produced two lightning strikes, although without the start of any new wildfires because lightning occurred over water in the Tasman Sea. However, it is well-documented that lightning activity from pyroCbs can start new ignitions (Dowdy et al., 2017).

As this research showed, the firestorm in the case of Dunalley wildfire was supported by favourable weather conditions which were associated with a highly unstable atmosphere and most probably triggered by the approach of the line of convergence in the hours prior to the cold front passage. In general, conditions supporting pyroCb or firestorm development are similar to those generating conventional thunderstorms (Tory and Kepert, 2021). The unstable atmospheric conditions are essential for pyrocumulus cloud formation, with the addition of a large heat source from the ground, in this case from the wildfire. The

difference between a firestorm and a regular storm in above-surface conditions is that lower levels of the atmosphere are drier, while the mid troposphere remains moist in both cases (e.g., Goens and Andrews, 1998; Cunningham and Reeder, 2009; Johnson et al., 2014). Such upper level conditions are found in the Dunalley case as well, where lower levels had relative humidity under 20 % while moist air with relative humidity of 70 % remained in the layer around 500 hPa.

Other significant upper-level conditions found in this case include the alignment of the northwesterly wind throughout the

troposphere and the strong low level jet. LLJ is related to a strong wind shear and turbulence in the atmospheric boundary layer and is a mesoscale/microscale feature that has been found in many large wildfire cases from USA, Europe and Australia to date (e.g., Byram, 1954; Vučetić et al., 2007; Čavlina Tomašević et al., 2022; Peace et al., 2022). LLJ influences wildfires through rapid changes in wind speed and direction, which would in turn influence fire spread rate and intensity especially when coupled with pyroconvection on a slope (Sharples et al., 2012). According to BARRA reanalysis, cross sections revealed

strengthening of wind at upper levels of the troposphere during the DUNALLEY 2 period. As previous research found that pyroCb in the case of Dunalley reached 13 km height (Ndalila et al., 2020), it remains to answer the question on how did pyroconvection manage to evolve to the extent it did in spite of strong wind above the wildfire ground. Further research is in plan to investigate the meteorological factors involved in pyroCb development in this case in detail.

One probable answer we can provide here is the enormous energy released from the wildfire which favoured pyroCb

development regardless of the surrounding conditions. Detailed analysis of fire intensity, which refers to the amount of energy generated from a fire (Keeley, 2008), found that the highest fire intensity value during the DUNALLEY 2 period reached 68, 517 kW m$^{-1}$ (Ndalila et al., 2018). As fire intensity ranges from 10 to 100,000 kW m$^{-1}$ (Byram, 1959), some studies categorize fire according to a certain value. For instance, fire intensity over 10,000 kW m$^{-1}$ and flame height over 11.5 m can define a fire as an extreme (e.g., Hirsch and Martell, 1996; Fernandes and Botelho, 2003). Another example categorizes fire intensity

between 7,000 and 70,000 kW m$^{-1}$ and flame height above 15 m as a complete crown fire with firestorm conditions in the case of fire intensities approaching the upper limit (Cheney, 1991). Thus, the Dunalley wildfire can be confirmed as an extreme wildfire with firestorm conditions. Previous study also adds that, together with extreme fire weather, maximum intensities found in the Dunalley case are caused by the abundance of dry fuel from *Dry Eucalyptus* forests situated northwest of Dunalley and, additionally, influenced by local topography which includes undulating terrain with mean slopes > 10° (Ndalila et al.,

490 2018).

4) The major change in the fire front direction was caused by the change in airflow which occurred with the cold front passage in the first hours of 5 January 2013 (the DUNALLEY 3 period). Crossing Tasmania, surface wind sharply switched from northwesterly to southwesterly and locally, near Dunalley, to southeasterly. This change influenced the wildfire by stopping the escalation in its activity and progression towards the southeast, over the Tasman Peninsula. While the cold front

at first affected the wind at lower levels, the upper-level northwesterly jet stream at first peaked, but eventually also decreased in speed and aligned to southwesterly direction on 6 January. The cold front passage brought a general decrease in wind speed and advection of cool and moist air from the southwest and caused the wildfire to extend back-burning northeast of the ignition




location, further inland, while the fire front along the Tasman Peninsula stopped its progression. Also, the change in conditions after the cold front passage took the fire into vegetation which would have higher moisture content (DPIPWE, 2020).

Earlier research showed that the intersection of hot continental air mass and cold maritime air over the Southern Ocean can intensify cold fronts as they approach the Australian continent (Luke and McArthur, 1978; Mills, 2002). Cold fronts are, as in the case of Dunalley wildfire, preceded by the prefrontal northerly wind and followed by southerly wind and advection of cool maritime air (Reeder and Smith, 1992; Reeder and Smith, 1998). When the maximum daytime temperature on the day after the frontal passage decreases by 12 to 17°C (12°C < ΔT < 17°C), it is considered a strong cold front; an extreme cold front is

when the difference is 17°C or greater (ΔT ≥ 17°C). Aforementioned research of 24 large wildfires in Australia in the period from 1962 to 2003 found 11 wildfires related to a strong cold front and 8 with an extreme cold front (Reeder, 2015; Mills, 2005; Long, 2006). Among those is the extreme cold front that influenced the Ash Wednesday wildfire in South Australia and Victoria in 1983. Almost all people killed during the Ash Wednesday wildfire died in just one hour after the cold front passed over the wildfire's area (Mills, 2005). Additionally, the most catastrophic Australian wildfire to date, the Black Saturday

wildfire in February 2009, was associated with an extreme cold front after which the daily maximum air temperature in Melbourne dropped from 46.4°C to 20.9°C (BoM, 2009; Cruz et al., 2012; Engel et al., 2013).

## 5.1 Concluding Remarks

To conclude, the presented meteorological analysis indicated that the Forcett-Dunalley wildfire occurred during an episode of severe weather conditions with the explosive pyroCb development triggered by the combination of the heat from the wildfire,

strong wind and atmospheric instability ahead of the cold front passage. The wildfire can be described as a combination of wind driven (DUNALLEY 1 and 3 periods) and buoyancy dominated wildfire (the DUNALLEY 2 period).

Here follows a short summary of fire weather drivers of the Forcettt-Dunalley wildfire:

- Climatically, the wildfire was preceded by months of warm and dry conditions which contributed to drying out fuels in the area.

- The first two days of the wildfire coincided with the peak of the extreme heatwave in Tasmania, which set 120-years record at Hobart station with maximum air temperature of 41.8°C.

- The FFDI of 112 reached catastrophic fire danger rating, which is considered a rarity in Tasmania.

- Surface synoptic conditions included a pronounced anticyclone situated northeast of Tasmania with approaching cold front from the west of the state. This synoptic pattern brought hot and dry conditions from the Australian mainland.

- The wildfire was carried by strong and gusty north-westerly wind which pushed the wildfire in southeast direction. Nothwest wind was the major driving force for most of the fire behaviour in this case.

- The explosive pyroconvection on the second day of the wildfire was triggered by unstable conditions in the atmosphere ahead of the cold front and enhanced by the abundance of dry vegetation and undulating local topography.

- The major ease in fire behaviour on the third day was caused by complete change in weather conditions after the

strong cold front passage, when wind shifted to southerly direction and air temperature dropped more than 20°C.

Occurrence of the Forcett-Dunalley wildfire is in agreement with the scientific consensus that the fire regime faces changes such as an increase in frequency, severity, intensity and size of wildfires, as well as changes in seasonality and type of wildfires (Collins et al., 2021). Specifically, there is a strong trend towards more extreme fire regime in the southeast of Australia, which is confirmed by multiple cases of FFDI values higher than anything previously recorded (Lucas, 2010; Enright and Fontaine,

2013). The research including a 67-year period of climate data (from 1950 to 2016) found evident changes of fire weather conditions in southern Australia with increase in frequency and extent of extreme wildfires together with earlier onset of fire seasons (Dowdy and Pepler, 2018). Moreover, numerous studies of future changes to FFDI in Australia without exception highlight the possibility of significant increase in fire danger, especially in the southeast (e.g., Cary, 2002; Hennessy et al., 2005; Pitman et al., 2007; Bradstock et al., 2009; Clarke et al., 2011; Dowdy et al., 2019), the region with the highest occurrence



of catastrophic wildfires in the recent decades (Teague et al., 2010). As mentioned before, studies from Tasmania found a springtime peak in fire danger, which appears approximately once in two years (Fox-Hughes, 2008), while future projections of mean sea level pressure related to elevated FFDI reveal increase in its frequency by the end of the 21st century (Fox-Hughes et al., 2014).

Only outputs of reanalysis data over Tasmania have been utilized here. An additional set of numerical simulations of higher
resolution is planned to use in future research. Future research will focus solely on weather conditions accompanying pyroconvection in order to give detailed insight into upper level conditions, including low level jet and conditions that favoured lightning activity in this case. The future study will also discuss the role that the line of convergence and approaching cold front might have had in this case of pyroconvection.

Meteorological analysis presented here leads to better understanding of fire weather conditions favourable for extreme fire
behaviour in Tasmania. Knowing the specific meteorological conditions related to extreme fire events such as the ones found in this research can improve fire weather forecasts for this region and other similar areas globally. Moreover, meteorological analyses of weather conditions during catastrophic wildfires and how they relate to fire behaviour (e.g., Charney et al., 2003; Coen et al., 2018; Brewer and Clements, 2020; Kartsios et al., 2020; Mass and Ovens, 2021; Čavlina Tomašević et al., 2022) can be used to educate and derive recommendations for fire management and to warn fire officials and the public on the
possibility of occurrence of such extreme events. Among other things, detailed systematic analyses of extreme wildfires can serve to increase awareness of potential for extreme events in today's more hazardous fire regime, which consequently can lead to improving prevention, decision making and planning.

**Code/Data availability.** Codes and datasets used in this study can be obtained from the corresponding author(s) upon
request.

**Author contribution.** ICT, PFH, KKWC, VV amd MTP conceptualized the study. KKWC, VV, PFH, MTP and PJB supervised the study. ICT and VV performed data analysis and visualization. ICT drafted the paper. ICT, PFH, KKWC, VV, JMS, PJB and MTP reviewed and edited the paper. KKWC, PFH, and PJB made English language edits.

**Competing interests.** The contact author has declared that none of the authors has any competing interests.





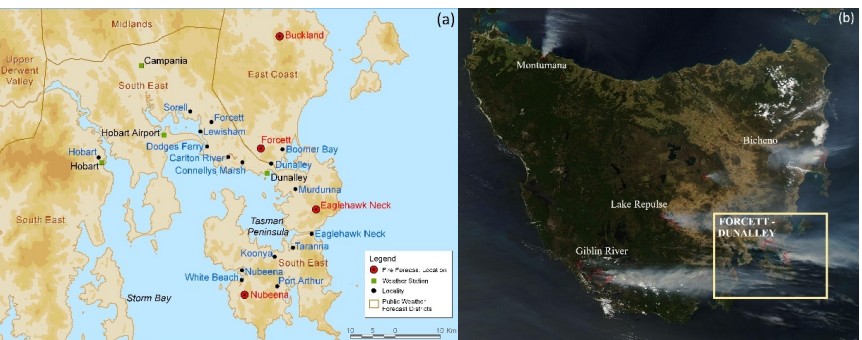

Figure 1. a) Weather station locations and localities in the Forcett-Dunalley area (BoM, 2013a) and b) satellite image of Tasmania with noted area of interest and smoke plumes visible from the MODIS Aqua satellite on 5 January 2013 (Image: MODIS Rapid Response, NASA).




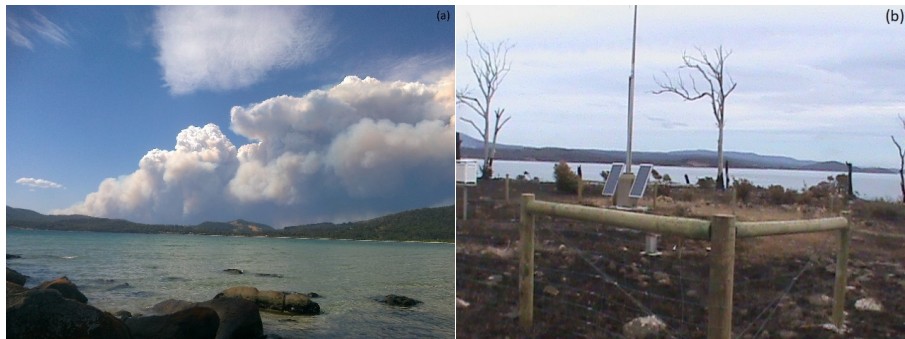

**Figure 2. a) Pyrocumulonimbus cloud during the Forcett-Dunalley wildfire at 16:00 AEDT (photographed by Janice James) and b)**
**weather station enclosure in Dunalley burnt in the wildfire between 16:19 AEDT and 18:06 AEDT, both on 4 January**
**2013 (photo from BoM, 2013).**



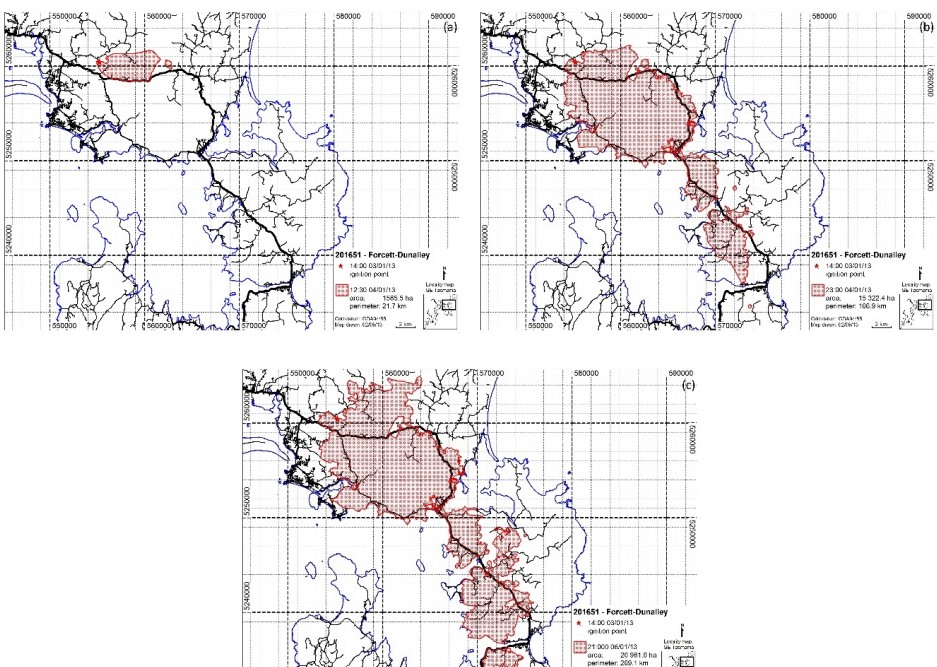


**Figure 3. Map of the Forcett-Dunalley wildfire with the final perimeter and three prominent progressions in growth. Burnt area a) by the end of DUNALLEY 1 period, b) by the end of DUNALLEY 2 period and c) by the end of DUNALLEY 3 period. Ignition location is noted as a red star. (Images credit: Dr Jonathon Marsden-Smedley, as part of work completed for the Bushfire and Natural Hazards Co-operative Research Centre.)**






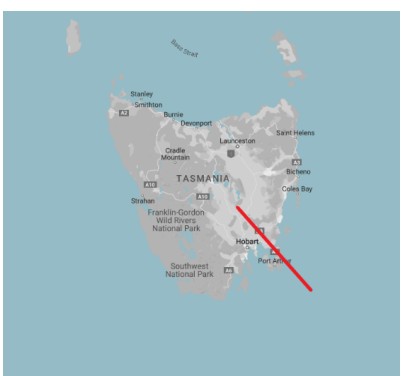

**Figure 4. Approximate BARRA-TA subdomain with the location of the defined cross section used here (Basic topography from https://tasmania.com/maps/tasmania/).**

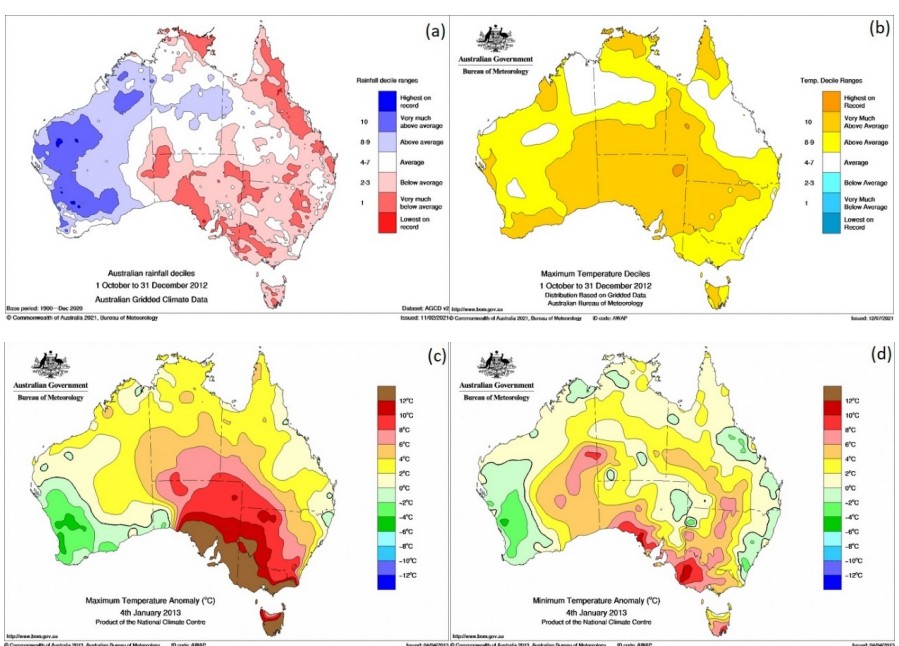


Figure 5. a) Rainfall deciles and b) maximum temperature deciles for Australia for the period from 1 October to 31 December 2012; and c) maximum air temperature anomaly and b) minimum air temperature anomaly for Australia for 4 January 2013 Anomalies are calculated against a standard reference period 1961 to 1990. (Maps from http://www.bom.gov.au/climate/maps/, last access: 29 September 2023).



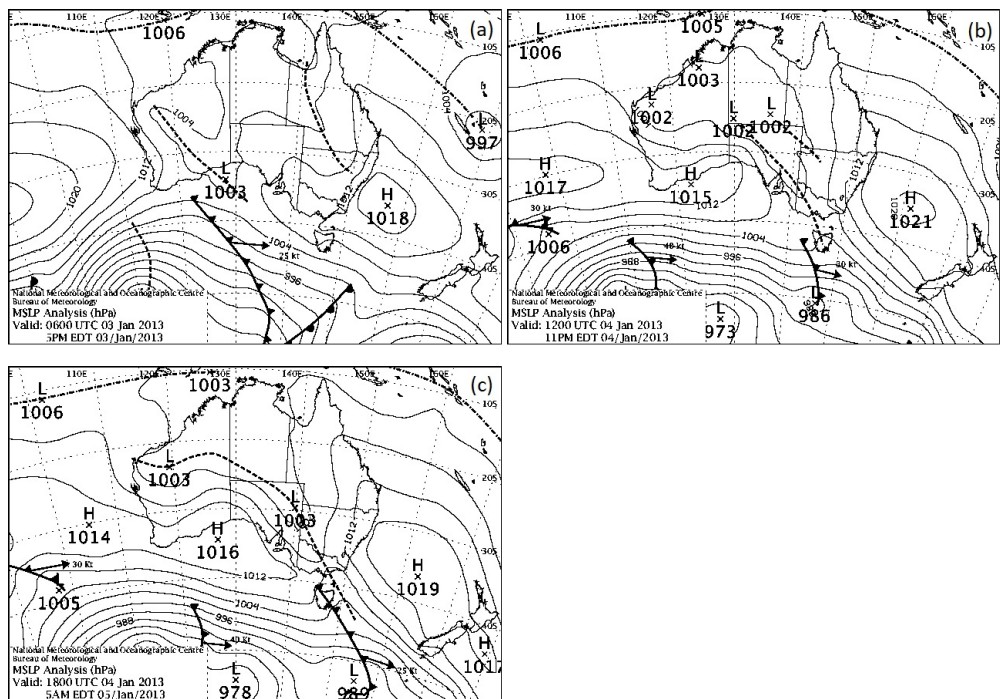


**Figure 6.** Mean sea level pressure (hPa) analysis for a) 17:00 AEDT on 3 January, b) 23:00 AEDT on 4 January, and c) 05:00 AEDT on 5 January 2013. Images: The National Meteorological and Oceanographic Centre (BoM).



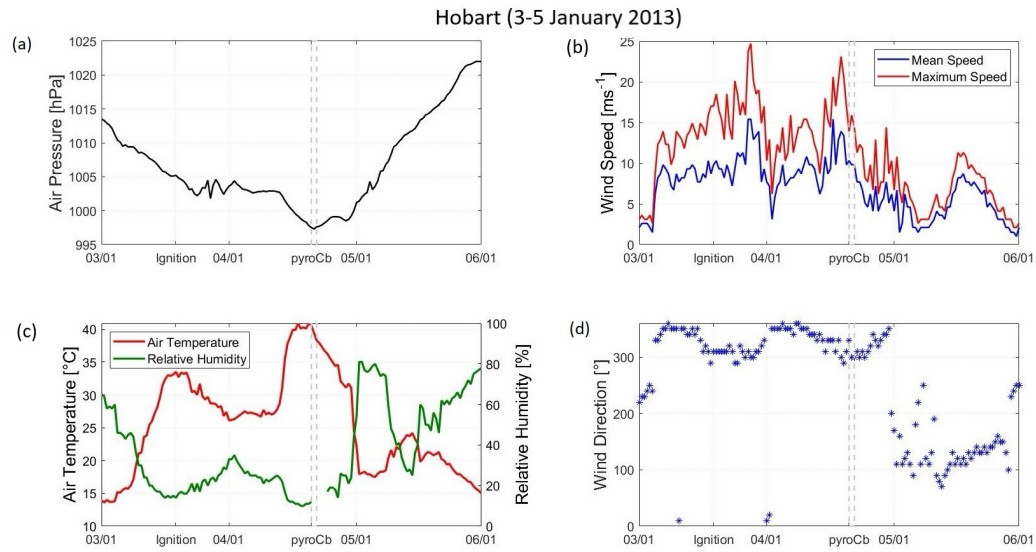


**Figure 7. Hobart automatic weather station 30-minute observations of a) air pressure (hPa), b) mean and maximum wind speed (ms$^{-1}$), c) air temperature (°C) and relative humidity (%), and d) wind speed direction (°) from 3 to 5 January 2013. Mean and maximum wind speeds are 10-minute values available every half hour. Missing values between 16:35 AEDT to 17:58 AEDT of relative humidity are due to corrupted measurements when the wet bulb reservoir dried out. Dashed grey lines indicate the peak activity of the pyroCb, between 15:30 AEDT and 16:00 AEDT on 4 January 2013 (the DUNALLEY 2 period).**



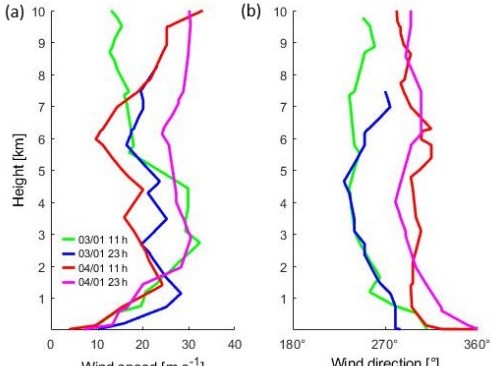


**Figure 8. Vertical profiles of a) wind speed (m s$^{-1}$) and b) wind direction (°) up to 10 km height, from Hobart Airport soundings at 11:00 AEDT (0000 UTC) and 23:00 AEDT (1200 UTC) on 3 January and 4 January 2013.**





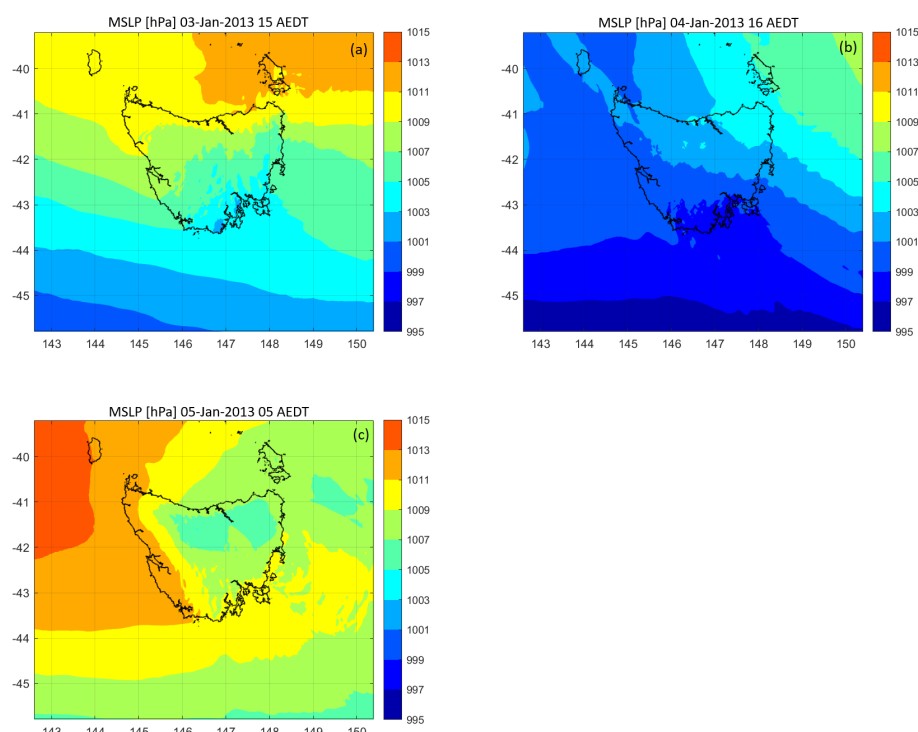

**Figure 9.** Mean sea level pressure (hPa) at a) 14:00 AEDT on 3 January, b) 16:00 AEDT on 4 January, and c) 05:00 AEDT on 5
January 2013 from BARRA reanalysis.

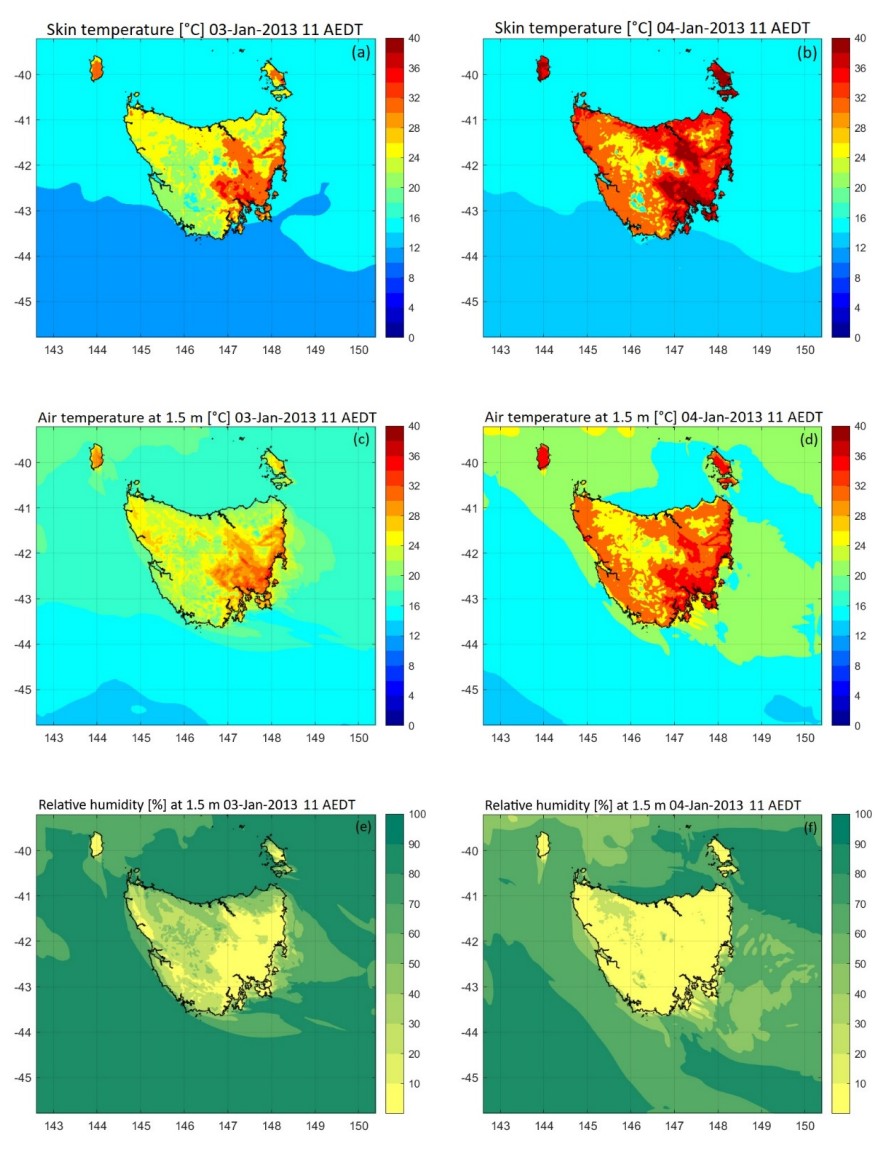


**Figure 10. Skin temperature (°C; top), 1.5-m air temperature (°C; middle) and relative humidity (%; bottom) at a), c) and e) 11:00 AEDT on 3 January 2013 and b), d) and f) 11:00 AEDT on 4 January 2013 from BARRA reanalysis.**






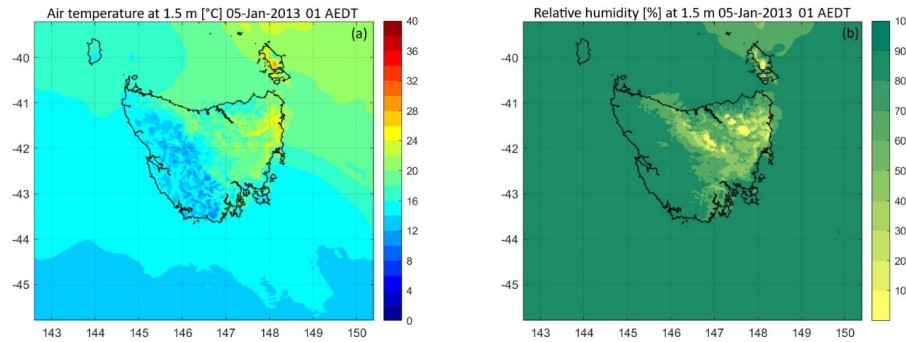

**Figure 11. a) Air temperature (°C) and b) relative humidity (%) at 01:00 AEDT on 5 January 2013 from BARRA reanalysis.**





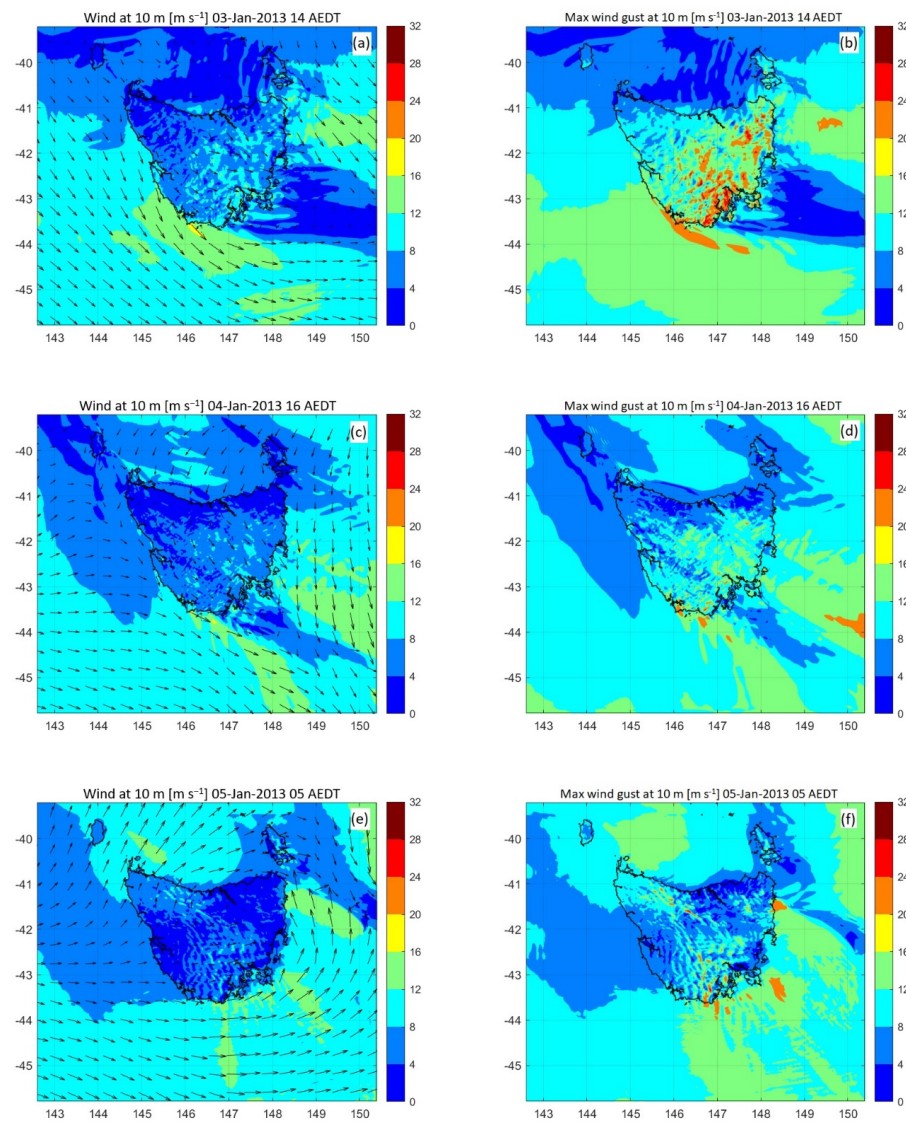

**Figure 12. Wind speed (m s⁻¹) and direction (left) and maximum wind gust (m s⁻¹; right) at 10 m at a) and b) 14:00 AEDT on 3 January 2013; c) and d) 16:00 AEDT on 4 January 2013, and e) and f) 05:00 AEDT on 5 January 2013 from BARRA reanalysis.**





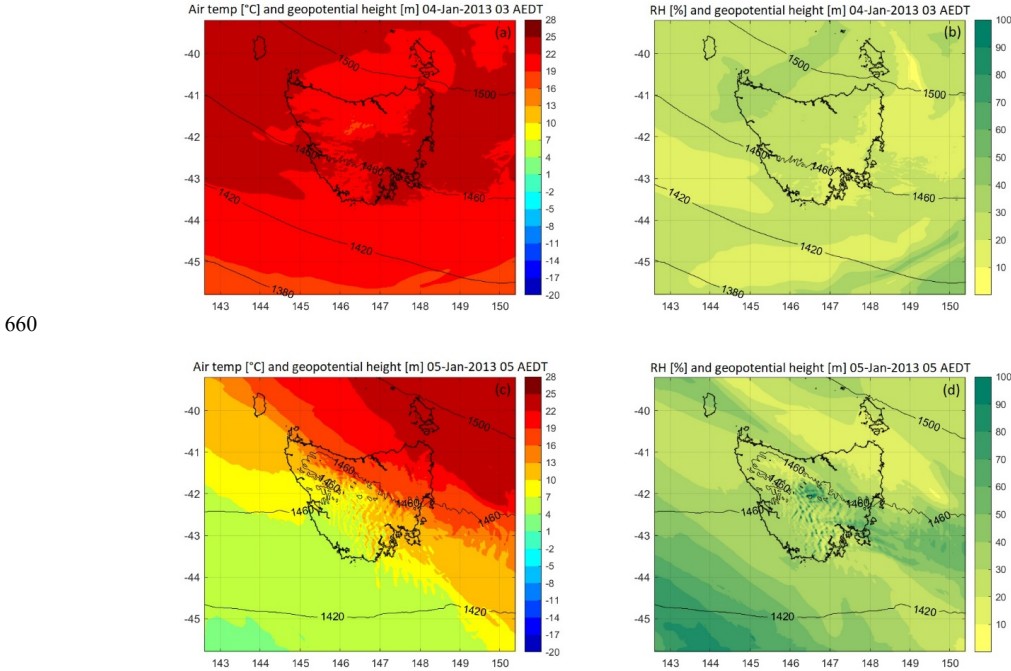

**Figure 13.** Air temperature (°C) and relative humidity (%) at 850 hPa at a) and b) 03:00 AEDT on 4 January 2013, and c) and d) 05:00 AEDT on 5 January 2013 from BARRA reanalysis.




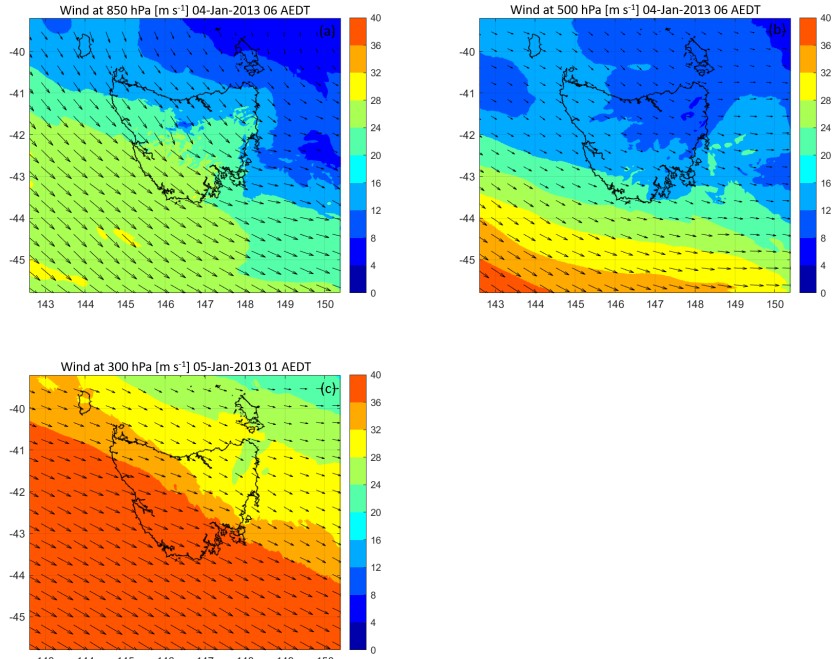

**Figure 14. Wind speed (m s⁻¹; coloured) and direction (array) at a) 850 hPa and b) 500 hPa at 06:00 AEDT on 4 January 2013, and c) 300 hPa at 01:00 AEDT on 5 January 2013 from BARRA reanalysis.**

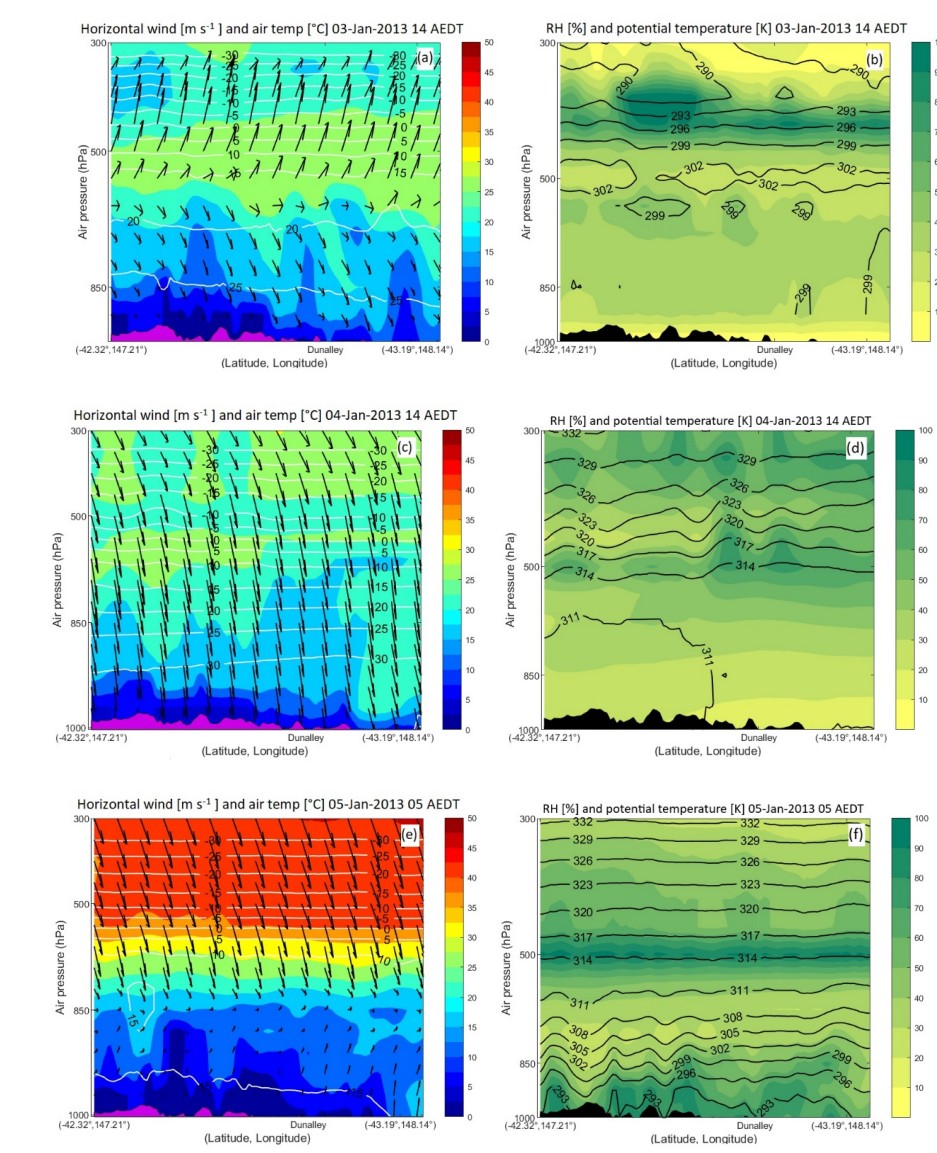

**Figure 15. Vertical cross sections of horizontal wind speed (m s⁻¹, colored) and wind vectors (arrows) and air temperature (°C), (left) and relative humidity (%, colored) and potential temperature (K; contour, right) at a) and b) 14:00 AEDT on 3 January 2013, c) and d) 14:00 AEDT on 4 January 2013 and e) and f) 05:00 AEDT on 5 January 2013 from BARRA reanalysis. The bottom violet (left panels) and black (right panels) area depicts the terrain. Air flow in each panel is from left to right.**



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
