# Peer review of "Meteorological Analysis of the Forcett-Dunalley Wildfire in 2013 in Tasmania, Australia"

_Natural Hazards and Earth System Sciences, 2023_

## Author Comment (AC1)

**Responses to Anonymous Referee (RC) #1 (nhess-2023-210)**

This is a case study describing some local weather observations and reanalysis data around the time of an extreme wildfire event in Tasmania. It is an interesting event, noting several studies publishing on this previously (including by some of these same authors https://www.mdpi.com/2073-4433/14/7/1076 and https://doi.org/10.5194/nhess-2019-354).

Some things could be enhanced in this study, including more supporting results for conclusions made. Examples for that include the following statements in the Abstract:

Response: Thanks for your detailed comments to our manuscript. In the revision, we have made the new findings clearer compared to previous studies on the same event, some of them by our co-authors. Point-by-point responses can be found as follow.

- The first statement in the Abstract after the data description says "antecedent climatic conditions in Tasmania included large increase in fuel load due to abundant rain one to two years before the event". But it is not shown in the results if there actually was a large increase in fuel load compared to normal. Details on what drives vegetation growth in that region are also not presented, such as if factors other than rain like soil temperature could potentially be relevant too.

  Response: Antecedent climate conditions have been analyzed in BoM (2013a), which we have quoted at the beginning of section 4.1. The same report has analyzed the soil dryness index near the end of 2012 to early 2013, which showed that the index was from normal to wet in that period and thus quite favorable for fuel to grow.

- The next sentence in the Abstract says "a low-level jet was directed downslope in southeast Tasmania to accelerate the fire spread" but it is hard to determine from the results presented that the jet caused accelerated fire spread.

  Response: This is based on our understanding of fire spread advected by the jet and in general downslope wind, such as in a European event and this Tasmania event analyzed in Tomašević et al. (2022, Atmosphere) and Tomašević et al. (2022, NHESS, doi:/10.5194/NHESS-22-3143-2022).

- The next sentence in the Abstract says "spotting of over 3 km was observed, and pyrocumulonimbus developed in this period with lightning up to 13 km from the fire". Spotting data and lightning data are not presented, or analysis to demonstrate pyrocumulonimbus occurrence (distinct from pyrocumulus).

Response: The spotting event to the Tasman Peninsula was documented in The Guardian (2013) and Marsden-Smedley (2014) reports and we estimated the distance to be about 3 km based on fire perimeter maps. Note that we also quoted 2.5 km in places in the manuscript and thus we have changed to 2.5 km for consistency.

Pyrocumulonimbus development with lightning up to 13 km was a fact from Ndalila et al. (2020). We have added citation to this reference.

- The final sentence in the Abstract says "Our analyses conclude that climatic conditions, synoptic patterns and mesoscale convective environment all contributed to this wildfire event", while noting previous studies have also discussed those types of aspects including the studies mentioned above on this event by some of these authors.

Response: In the revised abstract we have highlighted the advances of this study in apply the high-resolution BARRA-TA reanalysis to diagnose the mesoscale features of this event, which were not available in past studies that emphasized the climatic conditions and synoptic patterns.

In the data section of the manuscript, a key part of this is describing the FFDI and data used to calculate it. None of the figures show results for FFDI, but it could have been interesting to see examples leading up to and during FFDI being above 100, given that is discussed several times in the study (for example the 1-minute AWS data used in this study to calculate FFDI seems could have shown how variable the values are around their peaks). In various parts of the study the use of specific humidity, rather than relative humidity, would avoid ambiguity when examining moisture variability (given the temperature component of relative humidity).

Response: We have added the time series of FFDI as the new Fig. 6, which showed the increase to over 100 in the afternoon of the 4$^{th}$ January. The figure has been reproduced in the following.

[Figure]

Caption: Forest fire danger index (FFDI) at Hobart, Hobart airport, Dunalley and Campania based on 1-minute air temperature, dew point temperature and wind speed averaged over 10-minutes for the period from 05 AEDT to 23 AEDT on 4 January 2013. Values between 16:19 AEDT to 18:06 AEDT on 4 January 2013 at Dunalley station are officially omitted due to corrupted measurements when the wildfire affected instruments.

In general throughout, the study could be enhanced for accuracy. Examples for the section of text from lines 457-486:

- In this section of text it says "conditions supporting pyroCb or firestorm development are similar to those generating conventional thunderstorms (Tory and Kepert, 2021). The unstable atmospheric conditions are essential for pyrocumulus cloud formation". However, a key defining aspect of pyroCb occurrence is the influence of the fire on atmospheric conditions, which can sometimes occur in atmospheric conditions that are moderately or marginally stable, in contrast to atmospheric conditions in which conventional thunderstorms typically form.

Response: We agree that atmospheric conditions permitting pyrocumulonimbus development may be less unstable than is the case for conventional convection. We have modified the discussion to: "In general, conditions supporting pyroCb or firestorm development are similar to those generating conventional thunderstorms (Tory and Kepert, 2021). As noted by workers including Tory and Kepert (2021) however, the presence of a fire can modify low-level atmospheric stability, especially by heating, to increase the likelihood that deep moist convection will develop."

This section of text also seems to suggest the relative humidity around 500 hPa helped trigger pyroCb formation, but it is not clear how that occurred from the results presented (similarly also for lines 527-528).

Response: This is based on the understanding that mid-level moisture would enhance diabatic heating when updraft is initialized by the fire and promote further updraft to the upper level. A statement has been added to the paragraph and reference added to Tory et al. (2018) who discuss, among other things, entrainment into pyroCb plumes.

- The text here also says "fire intensity, which refers to the amount of energy generated" however, intensity here is (presumably) a rate of energy generated that might be integrated over time to give amount of energy.

Response: We have used kW/m as the unit of intensity and thus referred to the rate of energy generated. We have modified the first phrase to "fire intensity, which refers to the amount of energy generated per unit time".

- "Firestorm" is used here, and many times through the manuscript (or "fire storm" in some places). This could be kept consistent and clarified in definition for what it refers, as sometimes it seems to refer to fire-generated thunderstorms but not in other cases (such as lines 485-486 referring to a 1991 study not on pyroCbs).

Response: We refer to "firestorm" as a fire event with pyroCb development. We have stated this explicitly in its first appearance (section 2). Then in the discussion of the 1991 study without pyroCb, we have changed to "potential firestorm" as the intensity in that case reached an extreme value.

**Responses to Anonymous Referee (RC) #2 (nhess-2023-210)**

This study presents a meteorological analysis of the 2013 Forcett-Dunalley bushfire. The subject matter is within the scope of the NHESS journal, and both the title and the abstract accurately describe the contents of the study. Although the analysis from this study is applicable and useful, it is not completely clear if it represents a specific advance in knowledge. Other studies that analyze the fire and meteorological conditions for 2013 Forcett-Dunalley bushfire have already been published. The abstract and introduction sections can be revised to make it easier for the reader to identify which are the specific advances in knowledge offered by this study. Specific comments are listed below:

1. Line 23: The abstract states that spotting of over 3 km was observed, however subsequent sections reduce this distance to 2.5 km (lines 98 and 136).

   Response: The spotting distance was estimated based on fire perimeter maps and The Guardian (2013) report. However, in the revised manuscript we have removed the mention of this in the abstract.

2. Lines 67-68: The terminology is confusing. Technically speaking, a fire burns until it is extinguished. If the fire was not extinguished before March 20, it did not stop burning on January 18. Please use more specific terminology to describe what conditions changed for the fire on that date (e.g., was it controlled?)

   Response: The fire hadn't been completely controlled after 18 January. We have modified the sentence to "The Forcett-Dunalley wildfire had its major burning period from 3 January to 18 January 2013, however, it wasn't completely extinguished until 20 March 2013."

3. Lines 149-150: It is stated that 'the selected meteorological stations included Hobart […] and Hobart Airport'. Were these two stations the only ones that were considered for the analysis? If yes, why were the other stations shown in Fig. 1a not included (specially Dunalley)? If no, an improvement in the writing of this section can be beneficial for the readers.

   Response: There are two reasons that only the Hobart and Hobart Airport stations are included in this study. First, we focus on the BARRA reanalysis in this study and the two stations near the fire event suffice to show the general atmospheric conditions over southeast Tasmania (Figs. 7 and 8). Secondly, the Dunalley station data is relevant to the pyroconvection development, which we are analyzing and will be reported elsewhere. We have added a statement on this in the paragraph.

4. Line 136: The information provided does not explain how it was determined that the bushfire in the Tasman Peninsula was initiated by fire spotting originated in Eagle Neck, nor how the 2.5 km distance (or 3 km according to the abstract) was obtained. Moreover, while discussing fire spotting the readers are referred to Fig. 1a, but this figure focuses exclusively on presenting the location of weather stations and does not clearly identify in which location of the Tasman Peninsula the spotted bushfire was ignited.

   Response: We are sorry for the confusion. The referral to Fig. 1a was for the location of the Tasman Peninsula. The spotting over 2.5 km to the Peninsula was reported in The Guardian (2013) as well as our detailed examination of fire progression maps. Since these are indirect information, we have removed the statement about the spotting distance.

5. Line 175: The last two rows of Table 2 list '???' as the description of the difficulty of fire suppression of events during extreme and catastrophic FFDR. It is clear from the description for severe FFDR that suppression cannot be expected at that level of danger level and above, however it would be convenient to provide descriptions for the last two categories (e.g., fire suppression is considered impossible).

   Response: Added descriptions of fire danger for the highest two categories, based Australian Emergency Management Council guidelines (2009), plus a reference to that document. We have also noted that the McArthur system has now been replaced operationally, but that we use it in this manuscript for consistency with practices at the time of the fire.

6. Line 210: It is stated that the rainfall in 2009 and 2011 was 'very much above average', but no quantitative values are provided nor is it clear what the frequency and period over which the average was calculated are. Here (and previously in lines 207-208) the term 'very much' can be interpreted as subjective if no additional details are provided (e.g., a graph showing the yearly rainfall can provide important context for the readers).

   Response: The antecedent conditions were based on BoM (2013a). In particular, Fig. 3.2 in that report showed the positive rainfall anomaly in 2009 and 2011 especially over east Tasmania. We have referred to that figure in the revised manuscript and removed the subjective term 'very much' in our statement.

Technical corrections are listed below:

1. Lines 42-43: Percentages do not add up to 100%

Response: The percentages are from Nampak et al. (2021). There should be a round-up issue when they added up to 101%.

2. Line 395-426: These paragraphs repeat information that has already been presented in preceding sections of the study. This information can be omitted or summarized without compromising the message of the paper.

   Response: Repeated parts of the discussion have been removed and, following reviewer #3's recommendation, some additional contextual information has been added from these parts have been added to the results.

3. Line 470: 'low level jet' can be replaced with 'LLJ' to make it consisted with the rest of the document.

   Response: "low level jet" has been replaced by "LLJ".

**Responses to Anonymous Referee (RC) #3 (nhess-2023-210)**

This study aims to reconstruct the meteorological conditions leading up to and occurring during the Forcett-Dunalley wildfire. Although the subject matter seems appropriate for NHESS, I have some questions about the advances this work is making over previously published work and BoM reports. The work collects in one place, the different contributing weather factors to the fire and pyroCb storm, but I'm a bit unclear what original analysis the authors contributed here.

Response: The major advances of this work is to apply the high-resolution BARRA-TA reanalysis for diagnosing the meteorological conditions prior to the Dunalley fire. Thus, besides the climatic and synoptic conditions that have been analyzed in previous reports on the event, we have added mesoscale analysis on east Tasmania of the drivers of the fire progression and conditions for pyroconvection. This has been highlighted in the abstract for the benefit of readers.

The study uses mostly data from the BoM and the BARRA reanalysis, but there are places throughout the paper where more clarification is needed on the source of the observations. For example, in sec 2.1, line 98: how was flame height measured and how was spotting distance reported? Were these spot fires someone observed or reports of embers falling 2.5 km away from the front?

Response: The flame height and spotting distance was from The Guardian (2013) report.

And where are the numbers for spread rate coming from? Did the authors measure this themselves from satellite images or some other method? Or are those numbers from the BoM 2013 report? Is there any ability to quantify uncertainty on these measurements?

Response: Yes, the spread rates are from the BoM (2013a) report. We have added quotation in the statement. However, quantifying the uncertainty on these measurements is not feasible for this study.

Similarly for sec 2.2: what are the sources for some of these numbers? Are the spread rates coming from the BoM reports, or Ndalila 2018 or from analysis the authors did themselves? Reporting that "Upon arrival in Dunalley, the fire spread rate was 45 to 50m min-1" makes it unclear what information the authors are collecting from other sources and what they are calculating themselves.

Response: The spread rates are from the BoM (2013a) report. Quotation has been added to the statement.

Occasionally the authors make claims without quantifying anything specific to support those claims. For example, when talking about the wetter years in 2009 and 2011 that led to more dense vegetation, increasing the fuel load. These claims are plausible, but the authors should provide support since fuel load is a key parameter for wildfire growth. How much more rainfall was there in those years and can they quantify the fuel load changes over the years before the fire? Without backup, this section reads more as speculation, but could be strengthened greatly with some quantification since estimating fuel load would also be valuable to modelers.

Response: The antecedent conditions were based on BoM (2013a). In particular, Fig. 3.2 in that report showed the positive rainfall anomaly in 2009 and 2011 especially over east Tasmania. We have referred to that figure in the revised manuscript. The same report has also analyzed the soil dryness index, which is relevant to fuel growth, near the end of 2012 to early 2013. The index was from normal to wet in that period and thus quite favorable for fuel to grow.

Even in the FFDI calculation, is the assumption of a fixed value of 12.5 tons per hectare of fuel load appropriate given what was stated earlier about the denser vegetation?

Response: It is likely that fuel load exceeded the default value of 12.5 t/ha. However, forest fuel load is not routinely assessed for Australian fire danger calculations, and no quantitative estimates are available in this case. Since this analysis focusses on the mesoscale weather conditions contributing to the fire, it is outside the scope of the work to attempt an estimate of the actual fuel load. A brief comment has been added in the discussion of the FFDI noting that fuel load observations are not available, but that fuel load likely varied from the default value.

I also feel the paper could benefit from a reorganization, especially in the results & discussion section. The discussion section was very clear and helped contextualize the earlier portions of the results (which are very detailed play-by-play style reporting). I think intertwining the descriptive parts of the results with their relative discussion could make the importance of those observations much more clear and immediate.

Response: Sections of the first part of the discussion have been removed and inserted into the results, where they contextualize those results. See also our response to RC2 Technical Correction #2.

The paper could also use some language editing for clarity and grammar, and working hyperlinks for citations and figures and tables would be helpful.

Response: We have thoroughly edited the manuscript to improve clarity, grammar and the hyperlinks for citations, figures and tables.

Finally, the figures in this paper need to be either vectorized (where appropriate) or much higher resolution. Some of the labeling in the figures or the figures themselves can be hard to read even when zooming in. Color maps used could also be improved for clarity, for example in figure 8, since each line represents an evolution in time, a perceptually-uniform sequential colormap could be used to make the figure more readable, even in black & white. Other figures could also benefit from the use of perceptually uniform colormaps, instead of the default jet. Figures 9, 10, 11, 12, 13, 14, 15. Such colormaps not only improve readability but also make figures accessible to those with different forms of color vision deficiency.

Response: We have increased the resolutions of all our figures. In Figure 8, we have modified it to use a sequential colormap to illustrate the progressing time better. For the other figures, we think the use of the jet colormap has shown the separation of high and low values well, and thus we continue to apply it.